# Optimal Sensor Scheduling and Selection for Continuous-Discrete Kalman Filtering with Auxiliary Dynamics

**Mohamad Al Ahdab** [1] [2]  **John Leth** [1]  **Zheng-Hua Tan** [1] [2]

## Abstract

We study the Continuous-Discrete Kalman Filter (CD-KF) for State-Space Models (SSMs) where continuous-time dynamics are observed via multiple sensors with discrete, irregularly timed measurements. Our focus extends to scenarios in which the measurement process is coupled with the states of an auxiliary SSM. For instance, higher measurement rates may increase energy consumption or heat generation, while a sensor's accuracy can depend on its own spatial trajectory or that of the measured target. Each sensor thus carries distinct costs and constraints associated with its measurement rate and additional constraints and costs on the auxiliary state. We model measurement occurrences as independent Poisson processes with sensor-specific rates and derive an upper bound on the mean posterior covariance matrix of the CD-KF along the mean auxiliary state. The bound is continuously differentiable with respect to the measurement rates, which enables efficient gradient-based optimization. Exploiting this bound, we propose a finite-horizon optimal control framework to optimize measurement rates and auxiliary-state dynamics jointly. We further introduce a deterministic method for scheduling measurement times from the optimized rates. Empirical results in state-space filtering and dynamic temporal Gaussian process regression demonstrate that our approach achieves improved trade-offs between resource usage and estimation accuracy.

[1]Department of Electronic Systems, Aalborg University, Aalborg Øst 9220, Denmark. [2]Pioneer Centre for Artificial Intelligence, Copenhagen 1350, Denmark.. Correspondence to: Mohamad Al Ahdab <maah@es.aau.dk>.

*Proceedings of the $42^{nd}$ International Conference on Machine Learning*, Vancouver, Canada. PMLR 267, 2025. Copyright 2025 by the author(s).

## 1. Introduction

State-space models (SSMs) are fundamental tools for addressing sequential inference challenges in time-series forecasting, signal processing, and dynamic systems. A core task in SSMs is Bayesian filtering, which aims to compute the posterior distribution of latent states online given noisy observations. For linear systems driven by independently and identically distributed process noise with known dynamics, the Kalman filter (KF) is the optimal linear filter in mean squared error (Kalman, 1960). In the case of nonlinear dynamics, approximate inference methods such as the Extended Kalman Filter (EKF) and the Unscented Kalman Filter (UKF) are commonly used (Simon, 2006; Särkkä & Svensson, 2023). In this paper, we focus on the Continuous-Discrete Kalman Filter (CD-KF) setup, where continuous-time state evolution is observed via multiple sensors with discrete, and possibly irregularly timed, measurements. This choice is motivated by the fact that many real-world processes (e.g., blood pressure, ocean temperature, or radiation levels) evolve continuously but are only observed intermittently due to hardware constraints and operational considerations. Such formulation is especially relevant when selecting from multiple sensors operating under distinct costs, conditions, and constraints. Moreover, the measurement process is often coupled to auxiliary dynamics such as a sensor's temperature, spatial position, or stored energy, which affect measurement cost and quality. For example, consider a low-orbit satellite observing terrestrial phenomena (e.g., ocean temperature) using sensors whose accuracy depends on the satellite's orbital position. The satellite has two types of sensors: high-resolution sensors that are accurate during daylight periods, and radar-based sensors that are always available, but are energy-intensive and less accurate. Another example is a glucose monitoring setup where a diabetic patient tracks blood glucose via three different types of sensors: a continuous glucose monitor that provides noisy but frequently sampled measurements, intermittent finger-prick tests that offer higher accuracy but cause patient discomfort, and very accurate but infrequent and costly clinical blood tests. In these examples, the sensors differ not only in accuracy but also in their relation to auxiliary quantities such as energy consumption, monetary costs, and human discomfort. This means that adopting

a naive strategy of uniform high-rate sampling for all the sensors is undesirable and practically infeasible. Instead, measurement decisions must be scheduled to balance between minimizing the uncertainty in what they are measuring while simultaneously accounting for their interaction with dynamically changing auxiliary variables.

In the literature, most existing works do not consider the continuous-discrete setup. Additionally, scheduling measurements while simultaneously considering their interaction with an auxiliary state has not been explored directly in the literature. In this paper, we address this gap by formalizing the problem of optimal sensor scheduling for continuous-discrete SSMs with auxiliary dynamics and proposing an optimization scheme to solve it. Specifically, we model the arrival of the measurements from each sensor as a Poisson process with a time-varying rate. We then formulate an optimization problem that is continuously differentiable with respect to the measurement rates of each sensor and other relevant decision variables affecting the auxiliary dynamics (e.g, the trajectory of a vehicle carrying a sensor). Finally, we provide a deterministic strategy to select the measurement time instances for each sensor based on the optimized measurement rate for that sensor. Our contributions in this paper are summarized as follows:

1. We derive an upper bound on the mean posterior covariance matrix of the CD-KF conditioned on the mean auxiliary state. This bound is continuously differentiable with respect to the sensor-specific measurement rates, which enables efficient gradient-based optimization.

2. We propose the setup of a finite-horizon optimal control framework to jointly optimize measurement rates and other inputs that are relevant to the auxiliary dynamics.

3. We propose a deterministic method for obtaining measurement time events from the optimized rates. The method is based on minimizing the Wasserstein distance between the distribution of measurement events generated by the optimized Poisson rates and an empirical distribution determined by the expected number of measurements for each sensor.

## 2. Related Works

**Sensor Scheduling and Selection in Kalman Filtering:** The problem of optimizing sensor usage in Kalman filtering has been studied across diverse settings. Early work by (Ny et al., 2009) addressed continuous-time sensor management for Linear Time-Invariant (LTI) systems but assumed continuous measurements, a restrictive assumption for practical systems with discrete, asynchronous sensor observations. Subsequent studies focused on discrete-time formulations:

(Orihuela et al., 2014) developed periodic scheduling policies for discrete LTI systems, while (Marelli et al., 2019) introduced stochastic scheduling strategies under resource constraints. Optimal control perspectives have also been explored, such as the infinite-horizon formulation in (Zhao et al., 2014) and the finite-horizon networked control framework in (Ayan et al., 2020), which incorporated auxiliary network dynamics but retained a discrete-time LTI assumption. These works do not address the continuous-discrete setting with irregular measurements, nor do they consider the coupling between sensor scheduling and general auxiliary state dynamics (e.g., sensor's spatial trajectory and the available stored energy). Our work generalizes these approaches by unifying continuous-time state evolution with stochastic measurement scheduling via Poisson processes and integrating auxiliary state dynamics into the framework.

**Active Sensing:** Active sensing encompasses both sensor scheduling and trajectory optimization to maximize information gain. Recent advances leverage deep learning for sequential decision-making, for example, (Yoon et al., 2018) used recurrent neural networks to dynamically select medical tests based on patient history, while (Qin et al., 2024) proposed controlled neural ordinary differential equations to determine optimal measurement intervals in continuous-discrete settings. However, these methods focus on discrete classification tasks (e.g., disease diagnosis) rather than Bayesian state estimation. Recent work by (Napolitano et al., 2024) optimized for informative trajectories for Gaussian process (GP) regression in a model learning setup. However, these approaches do not consider the challenge of sensor scheduling and selection with general auxiliary state dynamics interacting with the sensor (e.g., the sensor's temperature affects its accuracy).

**Bayesian Optimization:** Bayesian Optimization (BO) has been widely adopted for optimizing expensive black-box functions, particularly in experimental design (Snoek et al., 2012). Classical BO methods rely on acquisition functions like Expected Improvement (Jones et al., 1998) or Upper Confidence Bound (Srinivas et al., 2012) to balance exploration and exploitation. Recent extensions integrate BO with reinforcement learning for sequential decision-making under uncertainty (Ling et al., 2016). BO has also been used to design experiments with mutual information in (Kleinegesse & Gutmann, 2020). These methods do not exploit known system dynamics or state-space structures. In contrast, our work leverages the analytic properties of the CD-KF to derive differentiable bounds on the estimation error (mean-squared error), which enables gradient-based optimization of measurement policies. Additionally, these approaches do not account for the influence of auxiliary dynamics on the measurements and their costs and constraints. Our formulation generalizes these settings by unifying continuous-time dynamics, stochastic measurement scheduling, and auxiliary

state constraints within a single optimal control framework.

To the best of our knowledge, the framework of scheduling sensors and measurements in a CD-KF setup with auxiliary dynamics has not been explored before. This work proposes this framework and presents some results on efficiently solving a class of this problem.

## 3. Notation

We denote by $\mathbb{S}_{\geq 0}^n$ ($\mathbb{S}_{>0}^n$) the cones of positive semi-definite (positive definite) $n \times n$ matrices. The Loewner order on $\mathbb{S}_{\geq 0}^n$ ($\mathbb{S}_{>0}^n$) is written as $A \preceq B$ ($A \prec B$), while element-wise inequalities are written as $\leq_e, <_e, \geq_e,$ and $>_e$. We write $I_n$ for the $n \times n$ identity matrix, and $\text{tr}(M)$ for the trace of a matrix $M$. The Dirac measure at $t_i$ is denoted $\delta_{t_i}$. The indicator function $\mathbf{1}_A : X \to \{0, 1\}$ is defined by

$$\mathbf{1}_A(x) := \begin{cases} 1, & x \in A, \\ 0, & x \notin A. \end{cases}$$

## 4. Background: Bayesian Filtering for SSMs

A (continuous-discrete stochastic) SSM is governed by:

$$dx = A(t)x dt + \sigma(t)dW, \quad x_0 \sim \mathcal{N}(\mu_0, \Sigma_0), \quad \text{(1a)}$$

$$y(t_i) = C(t_i)x(t_i) + v(t_i), \ v(t_i) \sim \mathcal{N}(0, R(t_i)), \quad \text{(1b)}$$

where $x \in \mathbb{R}^n$ is the state, $dW \in \mathbb{R}^m$ is a Wiener process, $A(t) \in \mathbb{R}^{n \times n}$ and $\sigma(t) \in \mathbb{R}^{n \times m}$ define the drift and diffusion matrices, $y(t_i) \in \mathbb{R}^q$ is a discrete-time measurement at time $t_i$ (here, $i$ indexes the measurement occasions, so $t_i$ denotes the $i$-th observation time), $v(t_i)$ is the measurement noise $v(t_i) \sim \mathcal{N}(0, R(t_i))$ with $R(t_i) \in \mathbb{S}_{>0}^q$ (independent and identically distributed), and $C(t_i) \in \mathbb{R}^{q \times n}$ is the output matrix.

For time instants $t_1 < t_2 < \cdots < t_i$, The Bayesian filtering problem involves sequentially obtaining $p(x(t_i) \mid y(t_1), \ldots, y(t_i))$. For linear-Gaussian SSMs, the CD-KF provides exact closed-form solutions for both the *filtering density* $p(x(t_i) \mid y(t_1), \ldots, y(t_i))$ and the *prediction density* $p(x(t) \mid y(t_1), \ldots, y(t_i))$ for $t > t_i$ (Jazwinski, 2013). Given a Gaussian prior $x(0) \sim \mathcal{N}(\mu_0, \Sigma_0)$ with $\mu_0 \in \mathbb{R}^n$ and $\Sigma_0 \in \mathbb{S}_{>0}^n$, the CD-KF follows the following steps:

**Prediction** ($t \in [t_{i-1}, t_i)$):

$$\frac{d\mu}{dt} = A(t)\mu, \quad \text{(2a)}$$

$$\frac{d\Sigma}{dt} = A(t)\Sigma + \Sigma A^\top(t) + \sigma(t)\sigma^\top(t), \quad \text{(2b)}$$

**Update** (at measurement $t_i$):

$$\mu(t_i) = \mu(t_i^-) + K(\Sigma(t_i^-), t_i)\left(y(t_i) - C(t_i)\mu(t_i^-)\right), \quad \text{(3a)}$$

$$\Sigma(t_i) = \left(I_n - K(\Sigma(t_i^-), t_i)C(t_i)\right)\Sigma(t_i^-), \quad \text{(3b)}$$

where

$$K(\Sigma(t_i^-), t_i) :=$$
$$\Sigma(t_i^-)C^\top(t_i)\left(C(t_i)\Sigma(t_i^-)C^\top(t_i) + R(t_i)\right)^{-1}, \quad \text{(4)}$$

is the Kalman gain. Let $N(t)$ be the total number of measurements up until time $t$, then the equation for the covariance $\Sigma$ in the CD-KF can be written compactly as

$$\frac{d\Sigma}{dt} = A(t)\Sigma + \Sigma A^\top(t) + \sigma(t)\sigma^\top(t)$$
$$- K(\Sigma, t)C(t)\Sigma \sum_{i=1}^{N(t)} \delta_{t_i}, \quad \text{(5)}$$

where $\delta_{t_i}$ is the dirac delta measure at $t_i$.

## 5. The Problem Setup

We consider a multi-sensor setup in which, at each measurement time, a sensor $s \in \{1, \ldots, S\}$ can be selected from $S$ available sensors. Additionally, we introduce an auxiliary state $\xi \in \mathbb{R}^{n_\xi}$ (e.g., representing the position of a vehicle and its energy storage), governed by the following dynamics:

$$\frac{d\xi_p}{dt} = f_p(\xi, u, t) + \sum_{s=1}^S g_s(\xi, u, t) \sum_{i=1}^{N_s(t)} \delta_{t_i^s}, \quad \text{(6a)}$$

$$\frac{d\xi_u}{dt} = f_u(\xi_u, u, t), \quad \text{(6b)}$$

where $\xi = [\xi_u \ \xi_p]^\top$. Here, $\xi_p \in \mathbb{R}^{n_p}$ denotes the "perturbed" part of $\xi$ affected by measurements, while $\xi_u \in \mathbb{R}^{n_\xi - n_p}$ represents the unperturbed part. The input signal $u \in \mathbb{R}^{m_\xi}$ (e.g., the velocity of a vehicle) influences the dynamics, and $N_s(t)$ represents the total number of measurements up to time $t$ for sensor $s$ with $t_i^s$ being the $i$-th measurement instant for that sensor. Note that the differential equation for $\xi_u$ depends only on $\xi_u$. Therefore, given an initial condition $\xi_u(0) = \xi_{u_0}$ and an input trajctory $u(t) \in \mathbb{R}^{m_\xi}$, then we can solve for $\xi_u(t)$ independantly from the measurement times. The functions $f_p, g_1, \ldots, g_S$ and $f_u$ are assumed to be Lipschitz continuous in $\xi$, $u$, and $t$.

Decomposing the auxiliary state into $\xi_p$ and $\xi_u$ is general and does not impose restrictions. For instance, if all of $\xi$ is perturbed by measurements, then $\xi = \xi_p$; conversely, if none of $\xi$ is perturbed, then $\xi = \xi_u$. This division becomes crucial later when analyzing the covariance structure of the KF. Often, such decomposition arises naturally. For example, $\xi_u$ may represent the kinematics or dynamics of a vehicle, which are typically independent of measurements, while $\xi_p$ might correspond to the stored energy of the vehicle, which depends on position and velocity (e.g., locations

for recharging or energy consumption rates at different terrains). Additionally, the stored energy can decrease with each measurement, and the consumed energy by each measurement event for sensor $s \in \{1, \ldots, S\}$ is characterized by $g_s$.

We now extend the SSM in (1a) to incorporate the auxiliary state $\xi$:

$$dx = A(\xi, t)x \, dt + \sigma(\xi, t) \, dW, \; x_0 \sim \mathcal{N}(\mu_0, \Sigma_0), \quad \text{(7a)}$$
$$y^s(t_i) = C_s(\xi(t_i), t_i)x(t_i) + v^s(\xi(t_i), t_i), \quad \text{(7b)}$$

where $y^s(t_i)$ is the measurement taken by sensor $s$ at time $t_i$, $C_s$ is the corresponding output matrix with the measurement noise being $v^s(\xi(t_i), t_i) \sim \mathcal{N}(0, R_s(\xi(t_i), t_i))$. Here, we also assume that $A, \sigma, C_1, \ldots, C_S$ and $R_1, \ldots, R_S$ are Lipshitz continuous in $\xi$ and $t$.

Our goal in this paper is to jointly optimize the measurement times for each sensor and the input $u$ within the context of a CD-KF.

In this paper, we will consider the following assumption for the dynamics of the auxiliary variables in (6).

**Assumption 5.1.** The functions $f_p$ and $g_s(\xi, u, t)$ are concave (convex) for all $s \in \{1, \ldots, S\}$ with respect to $\xi_p$ (in the elementwise or product order sense).

This concavity (convexity) assumption is essential for deriving upper (lower) bounds that lead to a tractable optimization problem for the state $\xi_p$. Importantly, it encompasses a broad class of systems, including all affine systems of the form $\alpha(\xi_u, u, t)\xi_p + \beta(\xi_u, u, t)$, which satisfy this assumption.

## 6. The Kalman Filter with Randomized Measurements

We now describe the randomized KF framework that forms the foundation for computing optimal measurement rates. Specifically, we model the arrival of measurements from each sensor $s \in \{1, \ldots, S\}$ as a Poisson process $N_s(t)$ with time-dependent intensity $\lambda_s(t)$. The composite measurement count is $N(t) = \sum_{s=1}^{S} N_s(t)$, and the randomized covariance matrix evolution for the CD-KF then becomes:

$$d\Sigma = \left(A(\xi, t)\Sigma + \Sigma A^\top(\xi, t) + \sigma(\xi, t)\sigma^\top(\xi, t)\right) dt$$
$$- \sum_{s=1}^{S} K_s(\Sigma, \xi, t)C_s(\xi, t)\Sigma \, dN_s, \quad \text{(8)}$$

where the Kalman gain $K_s(\Sigma, \xi, t)$ is given by:

$$K_s(\Sigma, \xi, t) :=$$
$$\Sigma C_s^\top(\xi, t) \left(C_s(\xi, t)\Sigma C_s^\top(\xi, t) + R_s(\xi, t)\right)^{-1}.$$

Similarly, the perturbed part of the auxiliary state becomes

$$d\xi_p = f_p(\xi, u, t) \, dt + \sum_{s=1}^{S} g_s(\xi, u, t) \, dN_s. \quad \text{(9)}$$

We now state the following two propositions, which enable us to formulate a deterministic optimization problem in the rates $\lambda_1, \ldots, \lambda_S$ and the input $u$. Throughout the rest of the paper, we will assume that the jump size functions satisfy the mean-square integrability condition (e.g., see Theorem 3.20 in (Hanson, 2007)).

**Proposition 6.1** (Covariance Matrix Bound). *Given initial conditions $\xi_0 \in \mathbb{R}^{n_\xi}$ and $\Sigma_0 \in \mathbb{S}_{>0}^n$, let $\xi(t)$ be the solution to (9) with $\xi(0) = \xi_0$ and $\Sigma(t)$ be the solution to (8) with $\Sigma(0) = \Sigma_0$. Let $\bar{\xi}(t) := \mathbb{E}[\xi(t)]$ with $\bar{\xi}(0) = \xi_0$ and $\bar{\Sigma}(t; \xi^*) := \mathbb{E}[\Sigma(t)|\xi(t) = \xi^*(t)]$ for some $\xi^*(t) \in \mathbb{R}^{n_\xi}$ with $\bar{\Sigma}(t; \xi^*) = \Sigma_0$. Then, the solution $\hat{\Sigma}(t)$ of*

$$\frac{d\hat{\Sigma}}{dt} = A(\xi^*, t)\hat{\Sigma} + \hat{\Sigma}A^\top(\xi^*, t) + \sigma(\xi^*, t)\sigma^\top(\xi^*, t)$$

$$- \sum_{s=1}^{S} \lambda_s(t)K_s(\xi^*, t)C_s(\xi^*, t)\hat{\Sigma}, \; \hat{\Sigma}(0) = \Sigma_0, \quad \text{(10)}$$

*satisfies $\bar{\Sigma}(t; \xi^*) \preceq \hat{\Sigma}(t), \forall t \geq 0$.*

**Proposition 6.2** (Auxiliary State Bound). *Let $\hat{\xi}(t) := [\hat{\xi}_p(t) \, \xi_u(t)]^\top \in \mathbb{R}^{n_\xi}$ be the solution to*

$$\frac{d\hat{\xi}_p}{dt} = f_p(\hat{\xi}, u, t) + \sum_{s=1}^{S} \lambda_s(t)g_s(\hat{\xi}, u, t), \quad \hat{\xi}(0) = \xi_0,$$
$$\text{(11)}$$

*where $\xi_u$ evolves according to (6b). If $f_p$ and $g_1, \ldots, g_S$ are concave (convex) with respect to $\xi_p$ (Assumption 5.1), then $\bar{\xi}(t) \leq_e \hat{\xi}(t) \left(\bar{\xi}(t) \geq_e \hat{\xi}(t)\right), \forall t \geq 0$.*

The proofs are provided in Appendix B.

*Remark* 6.3. If $A, \sigma, C_1, \ldots, C_S$, and $R_1, \ldots, R_S$ do not depend on $\xi_p$, then $\mathbb{E}[\Sigma(t)] = \mathbb{E}[\Sigma(t) | \xi(t) = \xi^*(t)]$. Moreover, if $f_p, g_1, \ldots, g_S$ are affine in $\xi_p$, then $\bar{\xi}_p(t) = \hat{\xi}_p(t)$.

## 7. The Optimal Control Problem

This section will utilize the bounds from Proposition 6.1 and Proposition 6.2 to formulate a general optimal control problem to calculate $u$ and the rates $\lambda_1, \ldots, \lambda_S$ for a horizon $[0, T]$. Consider that we have a continuously differentiable running cost $\mathcal{L} : \mathbb{R}^{n_\xi} \times \mathbb{S}_{>0}^n \times \mathcal{U} \times \mathbb{R}_{\geq 0}^S \to \mathbb{R}$ with a continuously differentiable terminal cost $\mathcal{L}_T : \mathbb{R}^{n_\xi} \times \mathbb{S}_{>0}^n \times \mathcal{U} \times \mathbb{R}_{\geq 0}^S \to \mathbb{R}$ which we desire to minimize (e.g., minimizing $\text{tr}(\hat{\Sigma})$). In addition to the costs, consider that we have $n_{c_r}$ continuously differentiable running constraints $\mathcal{C} : \mathbb{R}^{n_\xi} \times \mathbb{S}_{>0}^n \times \mathcal{U} \times \mathbb{R}_{\geq 0}^S \to \mathbb{R}^{n_{c_r}}$ together

with $n_{c_T}$ continuously differentiable terminal constraints $C_T : \mathbb{R}^{n_\xi} \times \mathbb{S}^n_{>0} \times \mathcal{U} \times \mathbb{R}^S_{\geq 0} \to \mathbb{R}^{n_{c_T}}$ (e.g, $\xi_p(T) \geq 0$ if $\xi_p(T)$ represent energy). We formulate the following general optimal control problem:

$$\min_{\substack{\lambda \geq_e 0,\ u \in \mathcal{U}, \\ \hat{\xi} \in \mathbb{R}^{n_\xi},\ \hat{\Sigma} \in \mathbb{S}^n_{>0}}} \int_0^T \mathcal{L}\left(\hat{\xi}, \hat{\Sigma}, u, \lambda\right) dt \tag{12a}$$
$$+ \mathcal{L}_T\left(\hat{\xi}(T), \hat{\Sigma}(T), u(T), \lambda(T)\right)$$

subject to

$$\frac{d\hat{\Sigma}}{dt} = A(\hat{\xi}, t)\hat{\Sigma} + \hat{\Sigma} A^\top(\hat{\xi}, t) + \sigma(\hat{\xi}, t)\sigma^\top(\hat{\xi}, t) \tag{12b}$$
$$- \sum_{s=1}^S \lambda_s(t) K_s(\hat{\xi}, t) C_s(\hat{\xi}, t)\hat{\Sigma},$$

$$\frac{d\hat{\xi}_p}{dt} = f_p(\hat{\xi}, u, t) + \sum_{s=1}^S \lambda_s(t) g_s(\hat{\xi}, u, t), \tag{12c}$$

$$\frac{d\xi_u}{dt} = f_u(\xi_u, u, t), \tag{12d}$$

$$\hat{\xi}(0) = \xi_0, \hat{\Sigma}(0) = \Sigma_0, \tag{12e}$$
$$\lambda(t) \geq_e 0,\ u(t) \in \mathcal{U},\ \forall t \in [0, T], \tag{12f}$$

$$\mathcal{C}\left(\hat{\xi}(t), \hat{\Sigma}(t), u(t), \lambda(t)\right) \leq_e 0,\ \forall t \in [0, T], \tag{12g}$$

$$\mathcal{C}_T\left(\hat{\xi}(T), \hat{\Sigma}(T), u(T), \lambda(T)\right) \leq_e 0, \tag{12h}$$

where $\mathcal{U} \subset \mathbb{R}^{m_\xi}$. Since $\hat{\Sigma}(t)$ and $\hat{\xi}_p(t)$ represent bounds on the true mean covariance and mean perturbed state, the cost functions $\mathcal{L}, \mathcal{L}_T$ and the constraints $\mathcal{C}, \mathcal{C}_T$ are assumed to be monotonically non-increasing (in the Loewner order) in $\hat{\Sigma}$, and monotonically non-decreasing (respectively, non-increasing) in $\hat{\xi}_p$ if $f_p$ and $g_1, \ldots, g_S$ are concave (respectively, convex) in $\hat{\xi}_p$ (element-wise)[1]. This ensures that the bounds from Propositions 6.1 and 6.2 provide meaningful constraints and costs. When $f_p$ and $g_s$ are *affine* in $\xi_p$, we have $\bar{\xi}_p(t) = \hat{\xi}_p(t)$ for all $t \geq 0$, so no additional monotonicity restrictions are required with respect to $\hat{\xi}_p$. Since we typically want to decrease or upper bound uncertainty, the requirement for monotonically increasing costs and constraints with respect to $\hat{\Sigma}$ is not restrictive. Note that one can also add slack variables to relax constraints and other optimization-related variables to (12). Furthermore, we assume the following:

**Assumption 7.1** (Feasibility). There exists at least one admissible pair $(u(\cdot), \lambda(\cdot))$ with a corresponding $\hat{\Sigma}(\cdot)$ and $\xi(\cdot)$ such that $\mathcal{C} \leq_e 0$ and $\mathcal{C}_T \leq_e 0$ are satisfied.

---

[1]We say for a differentiable function $f : \mathbb{R}^n \to \mathbb{R}^m$ that it is non-decreasing (non-increasing) if $\partial f_i / \partial x_j \geq 0$ $(\partial f_i / \partial x_j \leq 0)$ for $i \in \{1, \ldots, m\}$ and $j \in \{1, \ldots, n\}$.

Obtaining a closed-form solution to the problem in (12) is generally intractable. Thankfully, several numerical approaches can be utilized to obtain an approximate solution to the problem (Rao, 2009) (see Appendix D for a summary of these methods and the choice for the examples in this paper).

*Remark* 7.2. For the OCP in (12), we use $\hat{\xi}$ from Proposition 6.2 in the cost, in the constraints, and also for the dynamics of $\hat{\Sigma}$. By Proposition 6.1, $\hat{\Sigma}$ then upper-bounds the true covariance $\bar{\Sigma}(t; \hat{\xi})$. We distinguish three cases for our setup and the OCP in (12). The first case is when $f_p, g_1, \ldots, g_S$ are affine in $\xi_p$. In this situation, we have $\hat{\xi} = \bar{\xi}$ and the constraints and the costs in the OCP in (12) are for the exact mean trajectory $\bar{\xi}$. Additionally, $\hat{\Sigma}$ will be an upper-bound for the covariance matrix along the mean trajectory $\bar{\xi}$, and the costs and the constraints in the OCP will thus target the mean behaviour. If $f_p, g_1, \ldots, g_S$ are nonlinear but are concave (respectively, convex) in $\xi_p$ (Assumption 5.1), then $\hat{\xi}$ upper-bounds (respectively, lower-bounds) $\bar{\xi}$, so any monotonically non-decreasing (respectively, non-increasing) cost or constraint depending only on $\hat{\xi}$ remains meaningful. However, without further convexity/concavity assumptions on $A, \sigma, C_1, \ldots, C_S$ and $R_1, \ldots, R_S$ from (7), it need not follow that $\hat{\Sigma} = \bar{\Sigma}(t; \hat{\xi})$ bounds $\bar{\Sigma}(t; \bar{\xi})$. Nevertheless, if the dynamics are locally nearly linear (similar to the EKF mean approximation $\frac{d\bar{\xi}}{dt} \approx f_p(\bar{\xi}, u, t) + \sum_{s=1}^{N_s} \lambda_s(t) g_s(\bar{\xi}, u, t)$ (Simon, 2006; Särkkä & Svensson, 2023)), one finds $\hat{\Sigma} \approx \bar{\Sigma}(t; \bar{\xi})$, and thus, still approximately targets the mean behavior. Finally, in the fully nonlinear case violating Assumption 5.1, we can still treat $\hat{\xi}$ as a heuristic approximation for $\bar{\xi}$ without formal guarantees if the dynamics are locally nearly linear. Empirical results in Appendix H show that this approximation can yield acceptable performance in many scenarios.

*Remark* 7.3. While this paper focuses on an optimal control setup, Proposition 6.1 and Proposition 6.2 offer the opportunity to attempt different control strategies to compute $\lambda$ and $u$ depending on the specific problem of interest. For example, we can use control barrier function techniques (Ames et al., 2019) to find a feedback law for $\lambda$ and $u$ such that the set $\mathcal{S}_c := \{\Sigma \in \mathbb{S}^n_{>0} \mid \text{tr}(\Sigma) \leq c\}$ with $c > 0$ is forward-invariant.

# 8. Deterministic Selection of Measurement Times

After solving the OCP in (12) for the measurement rates $\lambda$ and inputs $u$, we would like to select the measurement times for each sensor $s \in \{1, \ldots, S\}$ within the horizon $[0, T]$ which are, loosely speaking, closely related to the average behavior of a Poisson process with rate $\lambda_s$. Relying on sampling the Poisson process to select measurement times is not suitable for a finite-horizon planning task as

we could sample a realization which is far from the average behaviour of the process. While this realization could provide a better solution, in terms of the cost and constraints in (12), than the average one, it could also provide a worse solution. Therefore, we aim to *deterministically* select the measurement times to be as close as possible to the average behavior according to a specific metric. To select deterministic measurement points $\bar{t}^s = (\bar{t}_1^s, \ldots, \bar{t}_{n_s}^s)$ for sensor $s$ with intensity rate $\lambda_s(t)$ over $[0, T]$, we cast the problem as one of optimal quantization. Define the normalized intensity measure

$$\mu_s(dt) \;=\; \frac{\lambda_s(t)}{\Lambda_s(T)}\, dt, \quad \text{where} \quad \Lambda_s(T) \;=\; \int_0^T \lambda_s(t)\, dt,$$

and let $\tau_s \sim \mu_s$ be the random variable that describes the time of a measurement from sensor $s$. Our goal is then to approximate $\mu_s$ by a discrete measure $\nu_s = \nu_{s,\bar{t}^s}$ given by:

$$\nu_s(dt) \;=\; \frac{1}{n_s} \sum_{i=1}^{n_s} \delta_{\bar{t}_i^s}(dt).$$

Denote $\tau_s^d \sim \nu_s$ to be the discrete random variable representing the time of measurement of sensor $s$ under $\nu_s$. We select $\bar{t}_s$ which minimizes the squared Wasserstein-2 distance,

$$W_2^2(\mu_s, \nu_s) \;=\; \inf_{\gamma \in \Gamma(\mu_s, \nu_s)} \int_{[0,T]^2} |t - t'|^2\, d\gamma(t, t'),$$

where $\Gamma(\mu_s, \nu_s)$ is the set of all couplings of $\mu_s$ and $\nu_s$. In one dimension, this distance simplifies to:

$$W_2^2(\mu_s, \nu_s) \;=\; \int_0^1 \left( F_s^{-1}(p) - G_s^{-1}(p) \right)^2 dp,$$

where $F_s$ and $G_s$ are the cumulative distribution functions (CDFs) of $\mu_s$ and $\nu_s$, respectively (Villani et al., 2009). In this case, we have $F_s(t) = \int_0^t \frac{\lambda_s(t)}{\Lambda_s(T)}\, dt = \frac{\Lambda_s(t)}{\Lambda_s(T)}$ and $G_s(t) = \frac{1}{n_s} \sum_{i=1}^{n_s} \mathbf{1}_{[\bar{t}_i^s, \infty)}(t)$.

**Proposition 8.1** (Optimal Quantization Points). *The points $\bar{t}_1^s, \ldots, \bar{t}_{n_s}^s$ that minimize $W_2^2(\mu_s, \nu_s)$ are the conditional centroids on $n_s$ intervals of equal probability under $\mu_s$. Specifically,*

$$\bar{t}_i^s \;=\; \mathbb{E}\left[ \tau_s \mid \tau_s \in [a_{i-1}^s, a_i^s] \right] \;=\; \frac{\int_{a_{i-1}^s}^{a_i^s} t\, \lambda_s(t)\, dt}{\int_{a_{i-1}^s}^{a_i^s} \lambda_s(t)\, dt}, \quad (13)$$

*where $a_i^s = F_s^{-1}\left( \frac{i}{n_s} \right)$ for $i = 1, \ldots, n_s - 1$, $a_0^s = 0$, and $a_{n_s}^s = T$. With this construction, $\nu_s$ matches the first moment of $\mu_s$ (i.e. $\mathbb{E}[\tau_s] = \mathbb{E}[\tau_s^d]$) and the variance difference is $\mathrm{Var}(\tau_s) - \mathrm{Var}(\tau_s^d) = W_2^2(\mu_s, \nu_s)$.*

The proof can be found in Appendix C. Note that $\Lambda_s(a_i^s) = \Lambda_s\left( F_s^{-1}(\frac{i}{n_s}) \right) = \Lambda_s\left( \Lambda_s^{-1}\left( \Lambda_s(T) \frac{i}{n_s} \right) \right) = \frac{\Lambda_s(T) i}{n_s}$ for all $i \in \{1, \ldots, n_s - 1\}$. In other words, in Propositions 8.1, we divide $\Lambda_s(T)$ into $n_s$ equal parts. This strategy distributes points more densely in regions where $\lambda_s(t)$ is large while preserving the mean of $\mu_s$ and minimizing second-order distortion in the Wasserstein-2 sense. The solution is unique and admits a closed-form expression, making it computationally efficient. Furthermore, minimizing the Wasserstein distance is equivalent to minimizing the quantization error $\mathbb{E}[\min_{t \in \bar{t}_s} ||\tau_s - t||^2]$, which is a classic result from optimal quantization theory for probability distributions (Graf & Luschgy, 2000). To match the average number of measurements, we select $n_s = \lfloor \Lambda_s(T) + 0.5 \rfloor$.

To clarify Proposition 8.1, assume for simplicity that $\Lambda_s(T)$ is an integer, then according to Proposition 8.1, we will partition $[0, T]$ into intervals such that $\Lambda_s(a_i^s) - \Lambda_s(a_{i-1}^s) = 1$ for $i \in \{1, \ldots, n_s\}$ if $n_s = \Lambda_s(T)$. This means that we partition $[0, T]$ into intervals wherein each one of them contains exactly one measurement on average. The location of that one measurement in each interval is taken to be the mean of $\frac{\lambda_s(t)}{\Lambda_s(a_i^s) - \Lambda_s(a_{i-1}^s)} = \lambda_s(t)$ over that interval. We select the measurement times for each sensor $s \in \{1, \ldots, S\}$ based on Proposition 8.1 independently.

# 9. Experiments

In many real-world applications (e.g., environmental monitoring (Alvear et al., 2017; Bird et al., 2018), and search-and-rescue (Waharte & Trigoni, 2010)), robots must operate under limited energy resources while gathering data in potentially hazardous environments. We consider such a scenario where a robot, constrained by its energy capacity, needs to collect measurements (e.g., pollutant concentration, air quality, temperature) using two onboard sensors for temporal GP regression [2].

## 9.1. Robot Model and Problem Setup

The state $\xi_u = [p_r, \theta_r]^\top \in \mathbb{R}^3$ represents the planner robot's position $p_r \in \mathbb{R}^2$ and its heading $\theta_r \in \mathbb{R}$. It evolves according to a unicycle model (see Appendix J for details). The input $u = [v, \omega]^\top \in \mathbb{R}^2$ consists of the heading velocity $v$ and the angular velocity $\omega$, both subject to physical bounds. The energy state $\xi_p = \eta \in \mathbb{R}$ evolves according to

$$\frac{d\eta}{dt} = c_e \exp\left( -r_e \|p_r - p_e\|^2 \right) - c_u v - c_u \omega - \sum_{s=1}^2 \sum_{i=1}^{N_s} c_s\, \delta_{t_i^s}, \quad (14)$$

---

[2] The code for generating the results, in addition to the corresponding animations and different examples, can be found on https://github.com/MOHAMMADZAHD93/When2measureKF

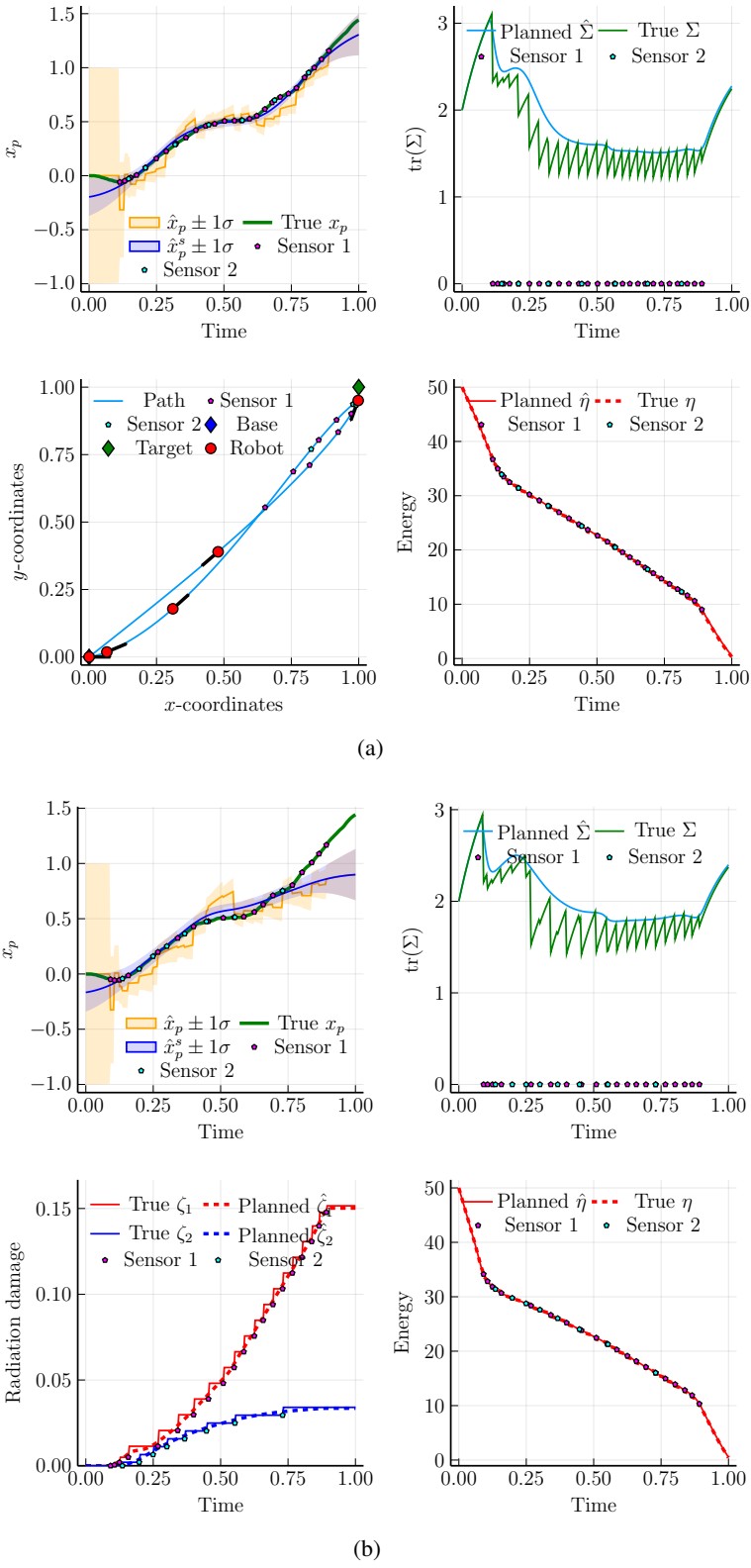

Figure 1. The planned proposed *mean* solution according to (12) and the true simulated solution based on the proposed measurement times selected according to (13) from the calculated rates $\lambda$ for (a) without radiation damage and (b) with radiation damage. $x_p$ represents the true process, $\hat{x}_p$ represents the KF estimate of it, and $\hat{x}_p^s$ represents the RTS estimate of it (the GP regression output).

where $c_u \geq c_1 \geq c_2 \geq 0$ are the energy costs of using the inputs and the sensors. The term $c_e \exp\left(-r_e\|p_r - p_b\|^2\right)$ (with $c_e, r_e \geq 0$) represents an energy charging source (e.g., solar) with its maximum charging rate at the base located at $p_b \in \mathbb{R}^2$.

The measurement noise covariance matrices depend on the robot–process distance:

$$R_s(p_r) = R_{s_{\max}} \exp\left(r_s\|p_r - p_p\|^2\right), \quad s \in \{1, 2\}, \quad (15)$$

where $R_{2_{\max}} > R_{1_{\max}} > 0$, $r_2 > r_1 > 0$, and $p_p \in \mathbb{R}^2$ is the location of the process. This setup implies that sensor 1 is more accurate but also consumes more energy than sensor 2. Starting at the base $p_b$ with initial energy $\xi_p(0) = \xi_0 \geq 0$, the robot must collect measurements at $p_p$ and return to $p_b$ by time $T \geq 0$ while ensuring that there is sufficient residual energy to continue operating. We employ the state-space representation of the Matérn kernel (degree 1) for the GPs (Särkkä & Hartikainen, 2012; Todescato et al., 2020). Kalman filtering and smoothing (Rauch-Tung-Striebel (RTS)) with this state-space representation is known to be equivalent to GP regression (see Appendix I for more details). The objective is to reduce uncertainties in the CD-KF estimate; hence, schedule measurements to minimize $\mathrm{tr}(\Sigma)$, where $\Sigma$ is the Kalman filter's covariance matrix for the estimated process (an upper bound on the smoothing covariance). In the OCP in (12), the running cost is

$$\mathcal{L}(\hat{\xi}, \hat{\Sigma}, u, \lambda, \varepsilon) = w_\Sigma \, \mathrm{tr}(\hat{\Sigma}) + w_\lambda \, \lambda^\top \lambda + w_u \, u^\top u + w_\varepsilon \, \varepsilon^2, \quad (16)$$

where $w_\varepsilon \gg w_\Sigma \geq w_\lambda \geq w_u \geq 0$, and the terminal cost is zero. To avoid large fluctuations in $\mathrm{tr}(\hat{\Sigma})$, we add the constraint $\mathrm{tr}(\hat{\Sigma}(t)) \leq c_\Sigma + \varepsilon(t)$ for $t \in [1/2, T]$, where $\varepsilon(t) \geq 0$ is a slack variable to ensure feasibility. We enforced the constraint on $\hat{\Sigma}$ only for $t \in [1/2, T]$ since requiring the trace of $\hat{\Sigma}$ to decrease rapidly can be restrictive and will always lead to large $\varepsilon$ at the beginning of the horizon. The running constraints $\mathcal{C}$ thus encodes the input bounds $\underline{u} \leq_e u \leq_e \bar{u}$, $\lambda \geq_e 0$, the trace constraint $\mathrm{tr}(\hat{\Sigma}) \leq c_\Sigma + \varepsilon$ (enforced for $t \in [1/2, T]$), $\varepsilon \geq 0$, and an energy lower bound $\hat{\eta} \geq c_\eta$ with $c_\eta \geq 0$. The terminal constraint $\mathcal{C}_T$ encodes that the robot should return to the base at the end of the horizon $p_r(T) = p_b$. Note that the dynamics of $\Sigma$ do not depend on the auxiliary state $\xi_p$. The results of the optimization and simulation for a horizon of $T = 1$ are illustrated in Figure 1(a).

## 9.2. Radioactive Environment Extension

We next examine a setting inspired by nuclear or chemical disaster scenarios, where the process of interest is located in a dangerously radioactive zone (Bird et al., 2018). Each measurement in this area degrades the sensors' accuracy over time. Specifically, the auxiliary state is now $\xi_p = [\eta \; \zeta_1 \; \zeta_2]^\top \in \mathbb{R}^3$, where $\zeta_1$ and $\zeta_2$ denote the accumulated

radiation damage on Sensors 1 and 2, respectively. The damage dynamics are

$$\frac{d\zeta_s}{dt} = \sum_{i=1}^{N_s} \gamma_i \exp\left(-r_\zeta \|p_r - p_p\|^2\right) \delta_{t_i^s}, \quad (17)$$

where $\gamma_1 > \gamma_2 \geq 0$, implying Sensor 1 is more susceptible to radiation damage than Sensor 2, and $r_\zeta > 0$. Consequently, the sensor covariance matrices become $R_s(\xi) = R_{s_{\max}} \exp\left(r_s\|p_r - p_p\|^2\right) \exp(r_s \zeta_s)$, $s \in \{1, 2\}$. Radiation damage thus exponentially increases the measurement noise of each sensor. Figure 1(b) presents results for this second scenario.

## 9.3. Comparisions

The approach used for the example jointly optimizes scheduling sensors while optimizing other inputs for the auxiliary dynamics. To our knowledge, there exist no methods that jointly optimize inputs for auxiliary dynamics together with scheduling sensors in a continuous-discrete setup. Therefore, we compare our approach with baseline strategies that utilize different measurement schemes while employing the optimized inputs for the auxiliary states. The baseline scheduling strategies are summarized as follows:

- Random: Discretize the horizon uniformly according to the number of discretization points for the OCP ($N_O$). Then, sample the measurement times for each sensor $s \in \{1, \ldots, S\}$ according to a Poisson process with constant rate $\lambda_s = N_O/S$.

- Greedy: Discretize the horizon into small time steps. Then, at each time step and for each sensor, compute a "score" equal to the immediate reduction in the trace of the predicted covariance minus the associated sensor costs and penalties on constraint violations. The measurement is then chosen according to the sensor with the highest positive score. If all the scores are negative, then we do not perform a measurement.

- M-Optimized: Instead of finding the measurement times according to the quantization rule in (13), we sample multiple realizations (10000 for the example in this paper) of measurement times according to the corresponding Poisson process with the optimized rates. Afterwards, we pick the measurement times corresponding to the realization with the minimum value of the cost function modified with penalties on constraint violations.

Table 1 summarizes the statistics for the comparisons where "Optimized" refers to our suggested approach.

*Table 1.* Trajectory statistics for the covariance trace, energy, and degradation over the horizon. The proposed approach in this paper is denoted as "Optimized".

| Covariance Trace | | | |
|---|---|---|---|
| Method | Mean | Std | Worst Case |
| Optimized | 1.90221 | 0.3406 | 2.94496 |
| M-Optimized | 1.89024 | 0.341106 | 2.94496 |
| Greedy | 2.60768 | 0.487677 | 3.21866 |
| Random | 2.27854 | 0.48601 | 2.96245 |
| Energy $\eta$ | | | |
| Method | Mean | Std | Worst Case |
| Optimized | 21.5435 | 10.1709 | 0.433792 |
| M-Optimized | 21.2173 | 10.5831 | -1.11621 |
| Greedy | -1.67343 | 14.0586 | -21.1662 |
| Random | -23.2507 | 33.7226 | -82.6662 |
| Degradation $(\zeta_1 + \zeta_2)$ | | | |
| Method | Mean | Std | Worst Case |
| Optimized | 0.0905529 | 0.0654335 | 0.185626 |
| M-Optimized | 0.0975802 | 0.0729591 | 0.213117 |
| Greedy | 0.193289 | 0.081316 | 0.232247 |
| Random | 0.873604 | 0.596903 | 1.68309 |

### 9.4. Discussion of the Results

From Figure 1, we observe that solving the optimal control problem in both scenarios yields a planned robot trajectory together with a corresponding measurement schedule for both of the sensors to track the measured process. We also remark from the figure that the planned upper bound $\hat{\Sigma}$ on the mean for $\Sigma$ nicely tracks the simulated true solutions for $\Sigma$ with measurement times calculated according to Proposition 8.1. We observe the same for $\eta$, $\zeta_1$, and $\zeta_2$. Additionally, we note that the trace of $\Sigma$ for the radioactive case is, on average, higher than that for the non-radioactive case, and the smoothing results (GP regression results) are visually worse for the radioactive case. This is expected since the radioactive environment case is more challenging. The results in Table 1 suggest that our method ("Optimized") outperforms the greedy and random scheduling approaches for the robot example. The results also demonstrate that our deterministic quantization provides similar results to "M-Optimized" without having to sample multiple realizations. Sampling multiple realizations can be computationally expensive and unrealistic since we do not have the real measurements to compute the corresponding cost for each realization. Overall, the results demonstrate the potential and feasibility of utilizing the proposed approach in the CD-KF setup with auxiliary dynamics for sensor scheduling and selection. See Appendix E for another example involving water quality

monitoring with fouling and active defouling. In addition, an example with spacecraft monitoring targets on Earth can be found in Appendix F. Moreover, Appendix G provides a discussion on how our approach can be generalized to uncertain and nonlinear SSMs with an example of a robot estimating the position and velocity of a moving target while attempting to track it. Finally, in Appendix H we provide examples for cases where the auxiliary dynamics do not satisfy Assumption 5.1.

## 10. Conclusion

We propose a novel methodology for sensor scheduling in SSMs that incorporates auxiliary state-space dynamics. By modeling the occurrence of sensor measurements through independent inhomogeneous Poisson processes with sensor-specific rates, we derive a continuously differentiable upper bound on the mean posterior covariance matrix of the CD-KF conditioned on the mean auxiliary state. This differentiability enables efficient gradient-based optimization of the sensor measurement rates. In addition, we formulate a finite-horizon optimal control problem that jointly optimizes these measurement rates with inputs for auxiliary dynamics. We further provide a quantization-based method for selecting deterministic sensor measurement times that closely match the mean behavior of Poisson processes with optimized measurement rates, thereby matching the actual sensor schedule with the intended resource–accuracy trade-offs. An Empirical evaluation in state-space filtering and dynamic GP regression confirms the feasibility and potential of this approach.

## Acknowledgements

This project is supported by the Pioneer Centre for Artificial Intelligence, Denmark.

## Impact Statement

This paper presents work whose goal is to advance the field of machine learning in general and Bayesian filtering in particular. There are many potential societal consequences of our work, none of which we feel must be specifically highlighted here.

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

# A. Convexity of the Kalman Update

The following lemma will be necessary to find the bound on Proposition 6.1.

**Lemma A.1.** *Let* $\Sigma \in \mathbb{S}^n_{>0}$, $C \in \mathbb{R}^{m \times n}$, *and* $R \in \mathbb{S}^m_{>0}$, *then the map* $\Sigma \mapsto \Sigma C^\top \left(C \Sigma C^\top + R\right)^{-1} C \Sigma$ *is convex.*

*Proof.* A map $f : \mathbb{S}^n_{>0} \to \mathbb{S}^n_{\geq 0}$ is convex if and only if its epigraph

$$\mathrm{epi}(f) \; = \; \left\{\, (\Sigma, X) \; \mid \; \Sigma \succ 0, \; X \succeq f(\Sigma)\right\}$$

is a convex set. Thus, we need to show that

$$X \; \succeq \; f(\Sigma) := \Sigma\, C^\top \left(C\, \Sigma\, C^\top + R\right)^{-1} C\, \Sigma$$

defines a convex constraint in $(\Sigma, X)$. By the Schur complement, $X \succeq f(\Sigma)$ holds if and only if

$$\begin{pmatrix} X & \Sigma C^\top \\ C\Sigma & C\Sigma C^\top + R \end{pmatrix} \succeq 0. \tag{18}$$

Since each block of this matrix depends *affinely* on the pair $(\Sigma, X)$, the set of all $(\Sigma, X)$ satisfying the above block-positivity is described by a linear matrix inequality. Since sets that are defined by linear matrix inequalities are convex, we conclude that

$$\mathrm{epi}(f) \; = \; \left\{ (\Sigma, X) \colon \Sigma \succ 0, \; \begin{pmatrix} X & \Sigma C^\top \\ C\Sigma & C\Sigma C^\top + R \end{pmatrix} \succeq 0 \right\}$$

is convex, which shows that $f(\Sigma)$ is convex. $\qquad\qquad\square$

Note that concavity and convexity are understood here in terms of the natural order structure associated with the notion of positive semidefiniteness (Loewner order).

# B. Proof of Proposition 6.1 and Proposition 6.2

We start by writing the conditional infinitesimals (Chapter 4 in (Hanson, 2007)) for (8) and (9)

$$\mathbb{E}\left[d\Sigma \mid \Sigma = \Sigma^*, \xi = \xi^*\right] = \left(A(\xi^*,t)\Sigma^* + \Sigma^* A(\xi^*,t)^\top + \sigma(\xi^*,t)\sigma(\xi^*,t)^\top - \sum_{s=1}^{N_s} \lambda_s(t) K_s(\Sigma^*,\xi^*,t) C_s(\xi^*,t)\Sigma^* \right) dt, \tag{19}$$

$$\mathbb{E}\left[d\xi \mid \xi = \xi^*\right] = \left( f_p(\xi^*,t) + \sum_{s=1}^{N_s} \lambda_s(t) g_s(\xi^*,t) \right) dt. \tag{20}$$

Let $\bar{\xi}(t) := \mathbb{E}[\xi(t)]$ then $d\bar{\xi} = \mathbb{E}[d\xi] = \mathbb{E}\left[\mathbb{E}\left[d\xi \mid \xi\right]\right]$ where the first equality follows by Fubini's theorem and the mean-square integrability condition, and the second equality by the tower property. Utilizing assumption 5.1 with Jensen's inequality, we therefore get

$$d\bar{\xi} = \mathbb{E}\left[ f_p(\xi,t) + \sum_{s=1}^{N_s} \lambda_s(t) g_s(\xi,t) \right] dt \leq_e (f_p(\bar{\xi},t) + \sum_{s=1}^{N_s} \lambda_s(t) g_s(\bar{\xi},t)) dt. \tag{21}$$

Letting $\frac{d\hat{\xi}}{dt} = f_p(\hat{\xi},t) + \sum_{s=1}^{N_s} \lambda_s(t) g_s(\hat{\xi},t)$, with $\hat{\xi}(0) = \bar{\xi}(0) = \xi_0 \in \mathbb{R}^{n_\xi}$, we conclude proposition 6.2 using the comparision theorem of ordinary differential equations (McNabb, 1986; Budincevic, 2010).

Similarly, denoting $\bar{\Sigma}(t; \xi^*) := \mathbb{E}[\Sigma(t) \mid \xi(t) = \xi(t)^*]$ and taking the expectation with respect to $\Sigma$ in (8), we get

$$d\bar{\Sigma}(t; \xi^*) :=$$

$$\left( A(\xi^*,t)\bar{\Sigma}(t;\xi^*) + \bar{\Sigma}(t;\xi^*)A(\xi^*,t)^\top + \sigma(\xi^*,t)\sigma(\xi^*,t)^\top - \sum_{s=1}^{N_s} \lambda_s(t)\mathbb{E}\left[K_s(\Sigma,\xi^*,t)C_s(\xi^*,t)\Sigma \mid \xi = \xi^*\right] \right) dt. \tag{22}$$

Using Lemma A.1, we can apply Jensen's inequality to obtain

$$\frac{d\bar{\Sigma}(t;\xi^*)}{dt} \preceq A(\xi^*,t)\bar{\Sigma}(t;\xi^*) + \bar{\Sigma}(t;\xi^*)A(\xi^*,t)^\top$$

$$+ \sigma(\xi^*,t)\sigma(\xi^*,t)^\top - \sum_{s=1}^{N_s} \lambda_s(t)K_s(\bar{\Sigma}(t;\xi^*),\xi^*,t)C_s(\xi^*,t)\bar{\Sigma}(t;\xi^*) := F(\bar{\Sigma},\xi^*,t) \quad (23)$$

Now let $\frac{d\hat{\Sigma}}{dt} = A(\xi^*,t)\hat{\Sigma} + \hat{\Sigma}A(\xi^*,t)^\top + \sigma(\xi^*,t)\sigma(\xi^*,t)^\top - \sum_{s=1}^{N_s}\lambda_s(t)K_s(\hat{\Sigma},\xi^*,t)C_s(\xi^*,t)\hat{\Sigma} = F(\hat{\Sigma},\xi^*,t)$ with $\hat{\Sigma}(0) = \bar{\Sigma}(0) = \Sigma_0 \in \mathbb{S}_{>0}^n$. By noting that $F$ is order-preserving in the covariance matrix ($\hat{\Sigma} \succeq \bar{\Sigma} \Rightarrow F(\hat{\Sigma},\xi^*,t) \succeq F(\bar{\Sigma},\xi^*,t)$), we conclude $\hat{\Sigma}(t) \succeq \bar{\Sigma}(t)$ for all $t \geq 0$ using the comparison theorem of ordinary differential equations. This concludes the bound in Proposition 6.1.

## C. Proof of Proposition 8.1

*Proof.* Since $\mu_s$ is supported on $[0,T]$ with CDF $F_s$, the quantile function

$$F_s^{-1}(p) = \inf\{t \in [0,T] : F_s(t) \geq p\}$$

partitions the interval into subintervals $[a_{i-1}^s, a_i^s]$ each having mass $\frac{1}{n_s}$ under $\mu_s$. The discrete measure $\nu_s$ has CDF $G_s$ with jumps at $\bar{t}_i^s$, so

$$G_s^{-1}(p) = \bar{t}_i^s \quad \text{for } p \in \left(\frac{i-1}{n_s}, \frac{i}{n_s}\right].$$

Hence,

$$W_2^2(\mu_s,\nu_s) = \int_0^1 \left(F_s^{-1}(p) - G_s^{-1}(p)\right)^2 dp = \min_{\bar{t}_1^s,\dots,\bar{t}_{n_s}^s} \sum_{i=1}^{n_s} \int_{\frac{i-1}{n_s}}^{\frac{i}{n_s}} \left(F_s^{-1}(p) - \bar{t}_i^s\right)^2 dp.$$

Taking derivatives with respect to each $\bar{t}_i^s$ and setting them to zero yields

$$\bar{t}_i^s = n_s \int_{\frac{i-1}{n_s}}^{\frac{i}{n_s}} F_s^{-1}(p)\, dp = \mathbb{E}\left[\tau_s \mid \tau_s \in [a_{i-1}^s, a_i^s]\right].$$

Uniqueness follows from the strict convexity of the squared-error objective. To see that $\mathbb{E}[\tau_s] = \mathbb{E}[\tau_s^d]$, observe that

$$\mathbb{E}[\tau_s] = \frac{1}{n_s}\sum_{i=1}^{n_s}\bar{t}_i^s = \int_0^T t\,\mu_s(dt) = \mathbb{E}[\tau_s^d],$$

since each $\bar{t}_i^s$ is the average of $\tau_s$ over the interval $[a_{i-1}^s, a_i^s]$. Finally, we have $\text{Var}(\tau_s) - \text{Var}(\tau_s^d) = W_2^2(\mu_s,\nu_s)$ which follows by the definition of $W_2^2(\mu_s,\nu_s)$. $\square$

## D. Numerical Solutions for Optimal Control Problems

There are several numerical methods to solve continuous-time OCPs. We list here the common methods used for solving them.

*Direct methods (Böhme & Frank, 2017)* work by discretizing the time axis and formulating the problem as a finite-dimensional Nonlinear Program (NLP). There are three common approaches for direct methods: direct single shooting, direct multiple shooting, and direct collocation.

In direct single shooting, the control trajectories (in our case $u(t)$ and $\lambda(t)$) are parametrized, for example, by polynomials, piecewise constant functions, or piecewise polynomials. The system dynamics are then integrated forward (using an ODE solver) to compute the objective and constraint values, with the NLP's decision variables corresponding to the control parameters. For direct multiple shooting, the time horizon is divided into smaller subintervals. In each subinterval, the control trajectory is parametrized as in the single shooting case. However, the dynamics are integrated separately for each subinterval instead of integrating them in one go for the entire horizon. The initial conditions for each subinterval for the

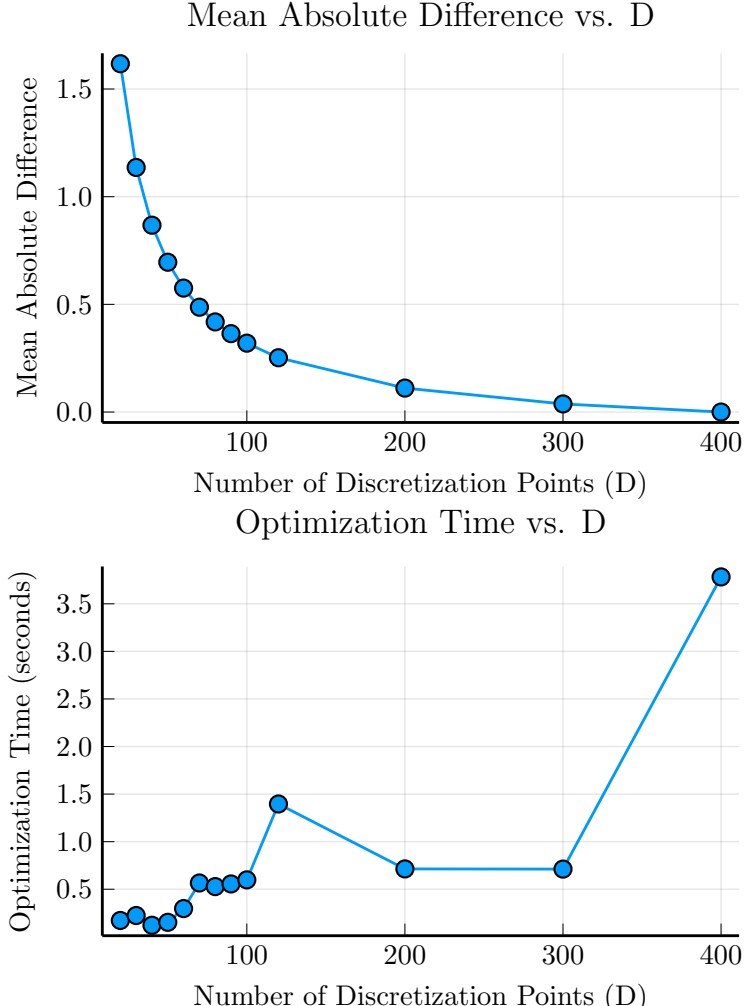

*Figure 2.* Mean absolute difference between all the trajectories and the one with $N_d = 400$. Computation times are reported for each $N_d$.

ODE solver are additional decision variables for the NLP with constraints ensuring the continuity conditions of the dynamics between the subintervals. Direct collocation (von Stryk, 1993; Bock & Plitt, 1984) follows a similar approach but also parametrizes the state trajectories together with the control trajectories.

Direct methods are well-suited for handling problems with many constraints and benefit from efficient off-the-shelf NLP solvers such as IPOPT (Wächter & Biegler, 2006). However, as the time grid becomes finer, it can lead to large-scale optimization problems.

*Indirect methods (Rao, 2009; Passenberg, 2012)* rely on necessary conditions for optimality (e.g., Pontryagin's Minimum Principle (Rozonoer, 1959)). One formulates the costate (or adjoint) equations and boundary conditions, then solves a Boundary Value Problem (BVP). While indirect methods can offer a deeper analytical insight into the problem, they can be challenging to implement for problems with many inequality constraints and nontrivial system dynamics. Additionally, numerical tools to solve BVP are sensitive to initial conditions.

In practice, the choice between direct and indirect methods often depends on problem size, constraint complexity, and the availability of good initial guesses or analytical insights.

For the examples in this paper, we choose a multiple shooting approach with forward Euler discretization and adaptive discretization steps. To ensure the positive definiteness of the covariance matrix in the NLP setup, we parametrized it with a

Cholesky decomposition. These choices are not limited, and one can, in principle, experiment with different choices based on the specific problem to solve.

It is important to note that discretization-based methods for OCPs with constraints over a horizon $t \in [T_1, T_2]$ may not strictly satisfy the constraints for all $t \in [T_1, T_2]$ due to numerical approximation errors. The choice of the discretization scheme inherently depends on the specific problem structure and its solution. While increasing the number of discretization points or adopting higher-order integration methods can improve accuracy, these enhancements often come with an increased computational cost. Figure D illustrates this trade-off in a two-sensor scheduling scenario for a two-dimensional CD-KF. We apply implicit Euler discretization with varying numbers of discretization points and quantify the resulting estimation error against a high-resolution reference (400 discretization points). Additionally, we record wall-clock runtimes on an Intel Core™ Ultra 7 155H @ 1.40 GHz device with 64 GB RAM. The optimization problems are formulated in Julia using the JuMP (Dunning et al., 2017) framework and solved with IPOPT (Wächter & Biegler, 2006). As the number of discretization points increases, the estimation accuracy improves, but at the cost of higher computation times.

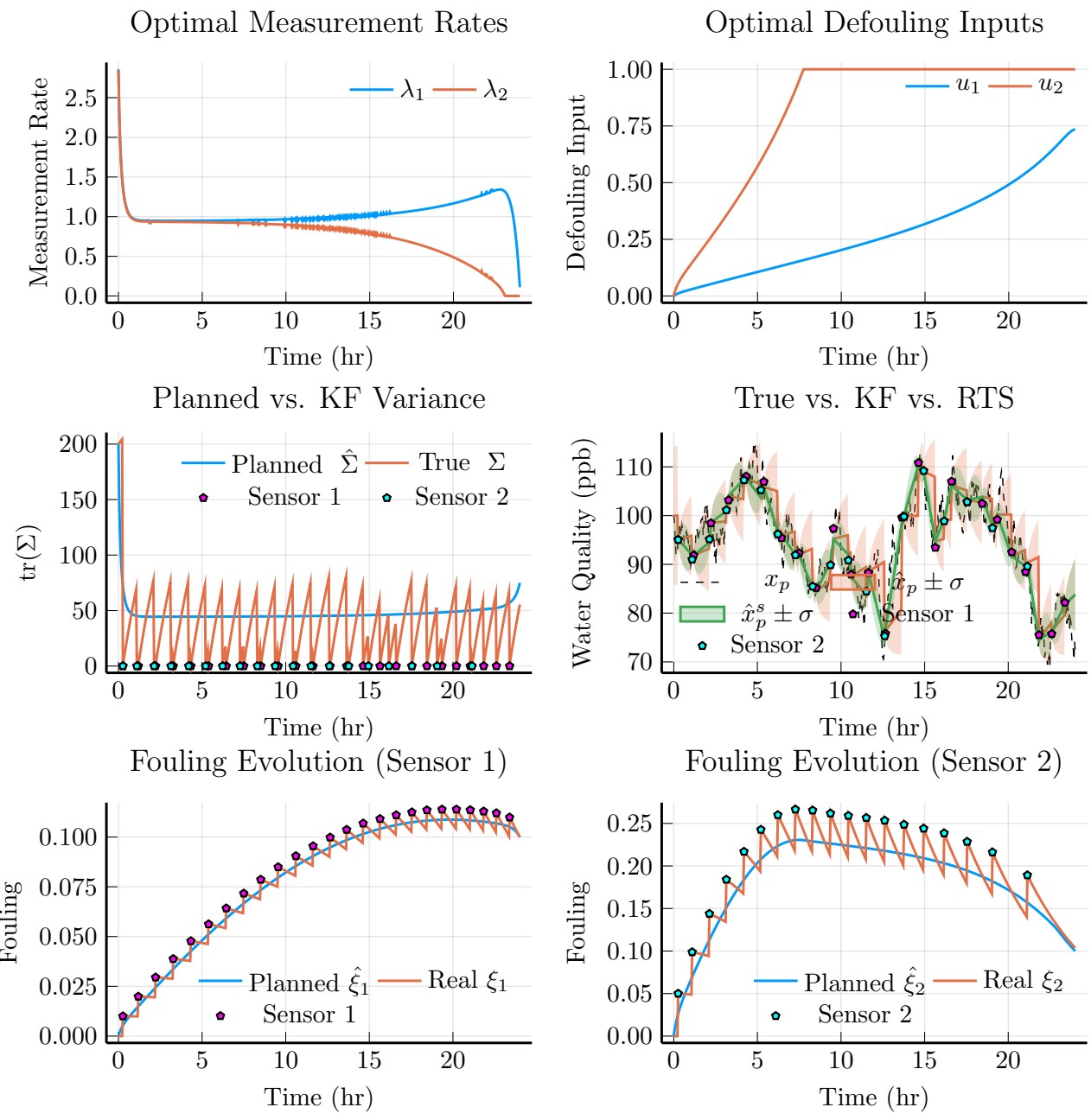

*Figure 3.* Water fouling example. The KF estimate is $\hat{x}_p$ and the RTS estimate is $\hat{x}_p^s$.

## E. Additional Example: Water Quality Monitoring with Fouling and Active Defouling

In many real-world water quality monitoring applications, sensors deployed in rivers, lakes, or coastal areas are prone to fouling due to the accumulation of biological, chemical, or sedimentary deposits. Fouling degrades sensor performance by increasing the measurement noise (Delauney et al., 2010). In addition, active defouling techniques (e.g., ultrasonic cleaning) have been developed to counteract fouling, although these methods incur their own operational costs (Delgado et al., 2021). In this example, we describe a model in which sensor scheduling is optimized jointly with active defouling control.

We assume that the water quality parameter evolves according to a linear model:

$$dx_p = -\kappa \left(x_p - x_{\text{amb}}\right) dt + \sigma \, dW, \quad x_p(0) \sim \mathcal{N}(\mu_0, \Sigma_0), \tag{24}$$

with parameters set to $\kappa > 0$, $x_{\text{amb}} > 0$, and $\sigma > 0$.

Each sensor provides a measurement of $x_p$ at discrete times where the noise covariance $R_s$ of each sensor $s \in \{1, 2\}$ depends on the fouling level via

$$R_s(\xi, t) = R_{s0} \, \exp\!\left(\lambda_s^f \, \xi_p^s(t)\right), \tag{25}$$

where $\xi_s^p$ is the fouling level for sensor $s$, $R_{10} > R_{20} > 0$, and $\lambda_1^f = \lambda_2^f > 0$. Higher fouling leads to a larger $R_s$ and, hence, lower measurement accuracy.

For each sensor $s$, the fouling level evolves as

$$\frac{d\xi_p^s}{dt} = -\left(\alpha_s^f + \gamma_s^f \, u_s(t)\right) \xi_p^s(t) + \sum_{i=1}^{N_s} \rho_s^f \delta_{t_i^s}, \ s \in \{1, 2\}, \tag{26}$$

where the parameters $\alpha_1^f = \alpha_2^f > 0$ represent the natural cleaning rate, $\rho_1^f, \rho_2^f > 0$ representing the parameters for the fouling increase per measurement with $\rho_2^f > \rho_1^f > 0$, $u_s(t) \geq 0$ represent active fouling control with rates $\gamma_1^f = \gamma_2^f > 0$. Remark that the second sensor can provide more accurate measurements ($R_{02} < R_{01}$) but is more prone to fouling with each measurement ($\rho_2^f > \rho_1^f$) due, for example, to an increased surface area. The goal is to schedule the measurements for a CD-KF while optimizing for the defouling inputs $u = [u_1 \, u_2]^\top$. We formulate an optimal control problem of the form in (12) for a horizon of $T = 24$ hr where the running cost is

$$\mathcal{L}(\hat{\xi}, \hat{\Sigma}, u, \lambda, \varepsilon) = w_\Sigma \operatorname{tr}(\hat{\Sigma}) + w_\lambda \, \lambda^\top \lambda + w_u \, u^\top u,$$

where $w_\Sigma \geq w_\lambda \geq w_{u_u} > 0$. The running constraints includes input bounds $\underline{u} \leq_e u \leq_e \bar{u}$ and an upper bound on the fouling for each sensor $\hat{\xi}_p^1, \hat{\xi}_p^2 \leq c_\xi$ with $c_\xi > 0$. Figure E shows the results. We can see from the results how the scheduling scheme brings the trace of the covariance matrix to a low level compared to its initial value. Additionally, we see that the planned measurements result in good estimates of the process. It is also worth noting how the OCP solution uses both the sensors equally in the beginning to reduce the trace of the covariance quickly. After that, the OCP solution reduces the rate of measurements for sensor 2 while saturating the defouling input to clean it. At the same time, it increases the rate of measurement for the first sensor while simultaneously increasing the defouling input at a faster rate.

## F. Additional Example: Spacecraft Monitoring Targets

We consider two spacecraft in circular low Earth orbit at an altitude of $500\,\text{km}$ separated by $\pi\,\text{rad}$. Each platform is tasked to monitor two processes on Earth's surface (for example, surface temperature and humidity). We model each target's local dynamics as an Ornstein Uhlenbeck process (equivalent to a Gaussian process with exponential kernel; see Appendix H):

$$dx_1 = A_1 \, x_1 \, dt + \sigma_1 \, dW_1,$$
$$dx_2 = A_2 \, x_2 \, dt + \sigma_2 \, dW_2,$$

where $A_1, A_2 < 0$ and $\sigma_1, \sigma_2 > 0$.

Each spacecraft has two distinct sensors. The first sensor on each spacecraft is a sensor whose accuracy depends on solar illumination. The second sensor on each spacecraft operates independently of solar illumination but requires higher energy per measurement. The accuracy of the first sensor with solar illumination is better than the second one, but it become worse in the absence of illumination. Additionally, daylight sensors are more prone to fail when used excessively compared to the other sensors. To capture these sensors in our framework, we denote the output matrices $C_1$ and $C_2$ for the first and second sensor in the first spacecraft, respectively, and $C_3$ and $C_4$ for the first and second sensor in the second spacecraft, respectively. We also denote the same for the corresponding measurement-noise variances $R_1, R_2, R_3, R_4$. Since each platform can only observe a ground target when it falls within its line of sight, we model the output matrices according to an angular visibility function:

$$C_s\!\left(\theta_{\text{sat}_i}, \theta_{\tau_j}\right) = \exp\!\left(-\left(\frac{\kappa_\delta \left(1 - \cos\!\left(\theta_{\text{sat}_i} - \theta_{\tau_j}\right)\right)}{1 - \cos(\delta)}\right)^4\right), \quad i \in \{1, 2\}, \ j \in \{1, 2\}, \ s \in \{1, 2, 3, 4\}.$$

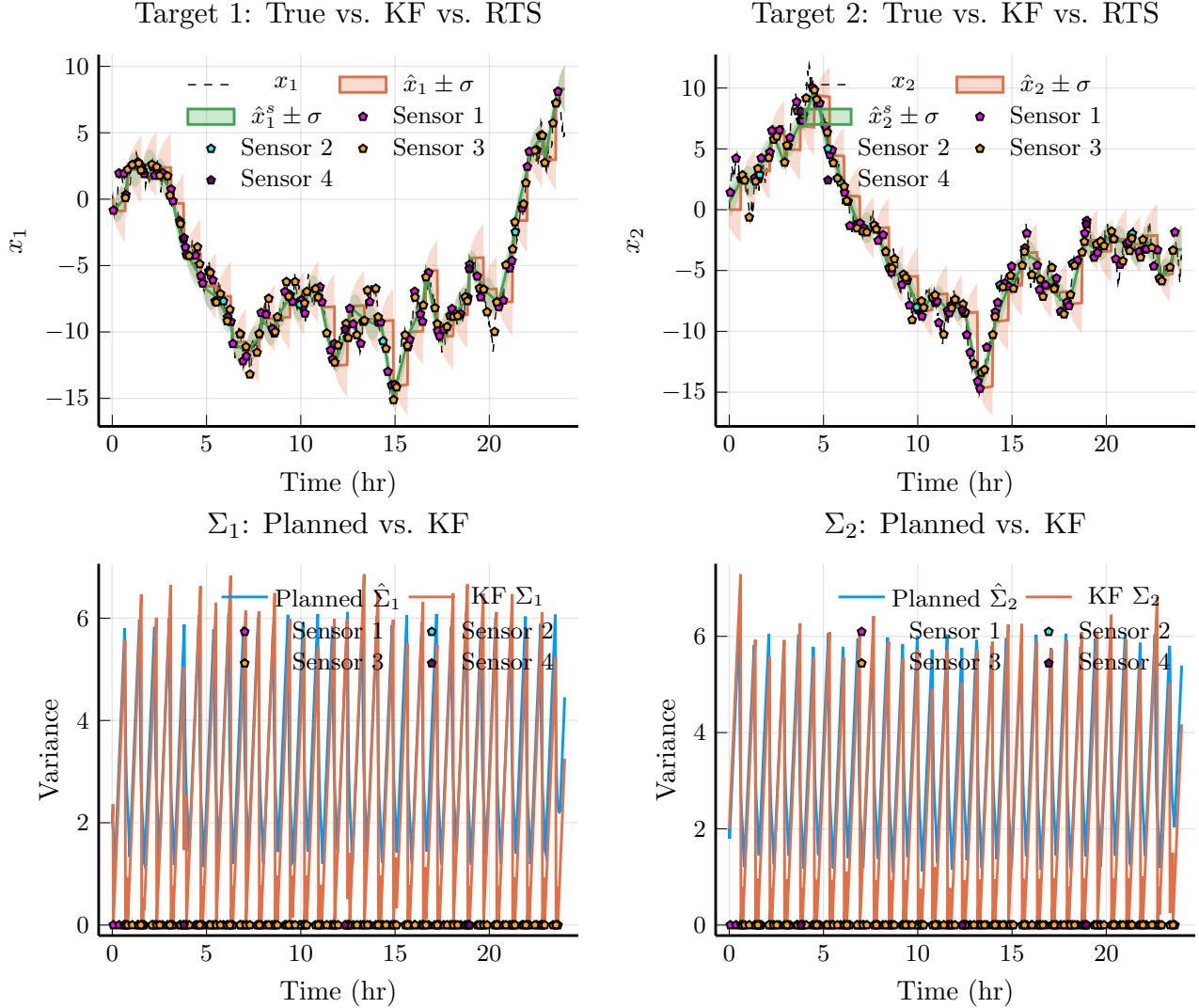

*Figure 4.* spacecraft example with the targeted processes. The KF estimates are $\hat{x}_1, \hat{x}_2$ and the RTS estimates are $\hat{x}_1^s, \hat{x}_2^s$. See an animation of the results on https://github.com/MOHAMMADZAHD93/When2measureKF.

Here, $\theta_{\text{sat}_i}$ denotes the angular position of the $i$-th spacecraft, $\theta_{\tau_j}$ denotes the geodetic longitude of the $j$-th surface target, $\delta$ is the half-cone angle defining the field of view, and $\kappa_\delta > 0$ is a shaping constant.

For the daylight-dependent sensors (Sensor 1 aboard spacecraft 1 and Sensor 3 aboard spacecraft 2), we model the instantaneous measurement variance as a smooth function of the solar illumination angle:

$$R_s\big(\theta_{\text{sat}_i}, \theta_\odot\big) = R_{\min}\, \rho_{\kappa_\odot}\big(\theta_{\text{sat}_i}, \theta_\odot\big) \;+\; R_{\max}\big[1 - \rho_{\kappa_\odot}\big(\theta_{\text{sat}_i}, \theta_\odot\big)\big], \quad i \in \{1,2\}, \; s \in \{1,3\},$$

where

$$\rho_\kappa(\theta, \theta') = \frac{1}{1 + \exp\big(-\kappa\, \cos(\theta - \theta')\big)},$$

where $\theta_\odot$ is the solar illumination angle, $R_{\max} > R_{\min} > 0$, and $\kappa_\odot > 0$ is a shaping constant determining how sharply the variance transitions from $R_{\min}$ (when the platform and sun are coaligned) to $R_{\max}$. For the solar independent sensors (sensor 2 and sensor 4), we fix the variance at a nominal value

$$R_2 = R_4 = R_\oplus,$$

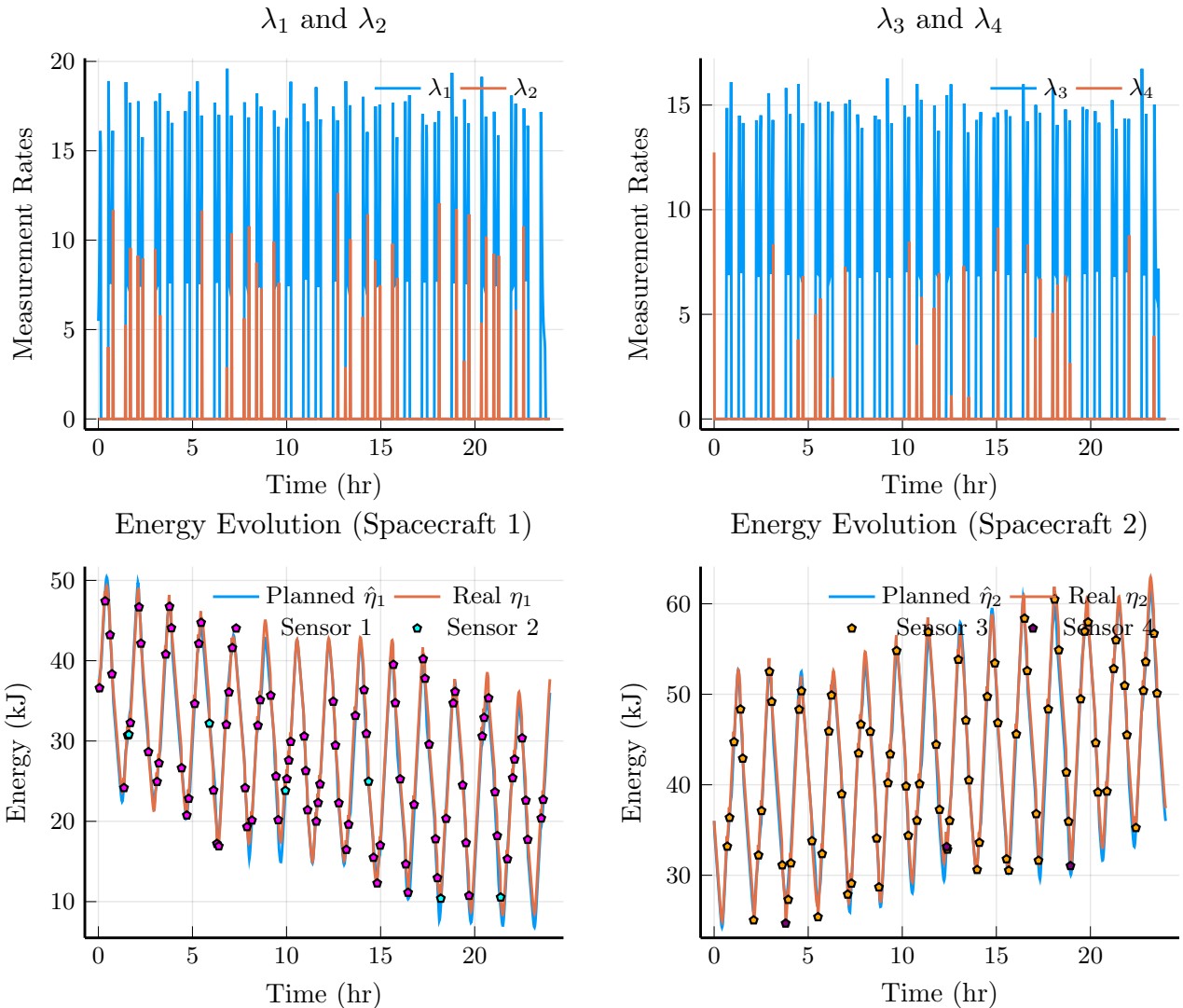

*Figure 5.* spacecraft example with measurement rates and the energy states. See an animation of the results on `https://github.com/MOHAMMADZAHD93/When2measureKF`.

chosen so that $R_{\min} < R_\oplus < R_{\max}$. In a more detailed analysis, one could distinguish Earth occultation of direct sunlight from simple misalignment, but here we adopt this smooth logistic model in order to keep the example concise.

Each spacecraft carries an onboard battery whose stored energy ($\eta_1$ for the first spacecraft and $\eta_2$ for the second one) evolves according to

$$\frac{d\eta_1}{dt} = -p_h + p_c\, \rho_{\kappa_\eta}\big(\theta_{\mathrm{sat}_1}, \theta_\odot\big)\, u_1 + \sum_{s=1}^{2} \sum_{i=1}^{N_s} c_s\, \delta_{t_i^s},$$

$$\frac{d\eta_2}{dt} = -p_h + p_c\, \rho_{\kappa_\eta}\big(\theta_{\mathrm{sat}_2}, \theta_\odot\big)\, u_2 + \sum_{s=3}^{4} \sum_{i=1}^{N_s} c_s\, \delta_{t_i^s},$$

where $p_h > 0$ is the nominal operating power of each spacecraft, $p_c > 0$ is the peak solar charging power, $\kappa_\eta > 0$ is a shaping constant, $u_1, u_2 \in [0, 1]$ are control variables controlling the charging of each battery such that the stored energy does not exceed an upper limit, and $c_s$ is the energy cost per measurement of sensor $s$, with $c_1 = c_3 > c_2 = c_4 > 0$.

We formulate an optimal control problem with a horizon of $T = 24$ hr. The running cost is

$$\mathcal{L}(\hat{\xi}, \hat{\Sigma}, u, \lambda, \varepsilon) = w_\Sigma (\hat{\Sigma}_1 + \hat{\Sigma}_2) + w_\lambda \lambda^\top Q \lambda,$$

where $\Sigma_1$ and $\Sigma_2$ are the variances of the CD-KF estimate for first target process and the second target process, respectively, and $w_\Sigma \geq w_\lambda \geq w_u > 0$, and $Q = \mathrm{diag}([10\ 1\ 10\ 1])$ used to increase the weights on Sensor 1 and Sensor 3 as they are more prone to fail with excessive use than Sensor 2 and Sensor 4. The running constraints includes input bounds $0 \leq_e u \leq_e 1$, and a bound on the maximum stored energy $\hat{\eta}_i \leq \eta_{\max}$, $i \in \{1, 2\}$. The results are shown in Figure F and Figure F. We see from F that the scheduling scheme manages to provide estimates tracking the processes nicely. We see an obvious periodic behaviour for the variances for each process. This periodicity is natural as both spacecraft passes over the target processes with periodically. We can observe this periodicity also in Figure F for both the scheduled measurement rates and the stored energy for each spacecraft.

## G. A Receding Horizon Framework for Unknown or Nonlinear Dynamics

In many practical scenarios, the dynamics that govern the process we wish to filter are either uncertain or nonlinear. One approach to extend our framework to handle these cases is a Receding Horizon (RH) formulation or a model predictive framework. RH approaches enjoy a rich literature showing that, by repeatedly solving a finite-horizon OCP with updated model information, one can retain provable closed-loop guarantees even when the prediction model is misspecified or learned online (Bemporad & Morari, 2007). Since our proposed approach in this paper is cast as an OCP, we can naturally extend it to an RH setup and utilize the existing approaches for robust or learning-based RH.

One way to extend our approach to an RH setup is to solve the OCP in (12) at time $t_l$ for a horizon $[t_l, T + t_l]$, obtain the measurement rates and the other possible control inputs over the horizon $[t_l, T + t_l]$, use (13) to find the time $t_{l+1}$ for the first sensor $s^*$ to measure within the interval $[t_l, T + t_l]$, apply the possible control for $t \in [t_l, t_{l+1}]$, measure with $s^*$ at $t_{l+1}$, update the KF estimates and the uncertain parameters of the model based on the measurement at time $t_{l+1}$, and re-solve the OCP for the horizon $[t_{l+1}, T + t_{l+1}]$ and repeat. In case of a nonlinear SSM, we can use the EKF (Simon, 2006; Särkkä & Svensson, 2023) for each OCP optimization. The EKF uses linearized dynamics for the covariance propagation, and we can update the linearized dynamics after each measurement between the finite-horizon OCPs. To illustrate this approach, we implemented a moving target tracking task in which a mobile robot with a unicycle model with an energy budget must localize and track a target whose true motion follows a constant turn rate velocity model perturbed by noise

$$
\begin{aligned}
dx_\tau &= v_\tau \cos \theta_\tau \, dt + \sigma_x \, dW_x, \\
dy_\tau &= v_\tau \sin \theta \, dt + \sigma_y \, dW_y, \\
dv_\tau &= \sigma_v \, dW_v, \\
d\theta_\tau &= \omega_\tau \, dt + \sigma_\theta \, dW_\theta, \\
d\omega_\tau &= \sigma_\omega \, dW_\omega.
\end{aligned}
\tag{27}
$$

where $p_\tau = [x_\tau, y_\tau]^\top$ is the position coordinates of the target, $\theta_\tau$ is the heading of the target, $v_\tau, \omega_\tau$ are the heading linear and angular velocity of the target, respectively, and $\sigma_x, \sigma_y, \sigma_v, \sigma_\theta, \sigma_\omega > 0$ with $W_x, W_y, W_v, W_\theta$ and $W_\omega$ being independent Wiener processes. The OCP and the CD-KF assume a constant velocity model for the target with random perturbations on the velocity only

$$
\begin{aligned}
dx &= \tilde{v}_x \, dt \\
dy &= \tilde{v}_y \, dt \\
d\tilde{v}_x &= \sigma_{v_x} \, dW_{v_x}, \\
d\tilde{v}_y &= \sigma_{v_y} \, dW_{v_y},
\end{aligned}
\tag{28}
$$

where $\tilde{p}_\tau = [x, y]^\top$ is the position coordinates of the target, $\tilde{v}_x, \tilde{v}_y$ are the linear velocities on the horizontal and vertical axis, respectively, and $\sigma_{v_x}, \sigma_{v_y} > 0$ with $W_{v_x}$ and $W_{v_y}$ being independent Wiener processes. We assume we have two sensors for the position of the target and that the measurement noise covariance matrices are given as the ones in (15) with $p_p$ replaced with $p_\tau$ for simulation and the estimated $\tilde{p}_\tau$ in the OCP. Additionally, the energy of the robot follows the same dynamics in (14) but with a constant charging rate that does not depend on the position of the robot (more suitable for target tracking). We used an RH setup to plan for the measurements and the robot's velocities for a horizon of $T = 0.05$ where for each OCP optimization, we use the current estimate for the target according to the constant velocity model (28) instead of

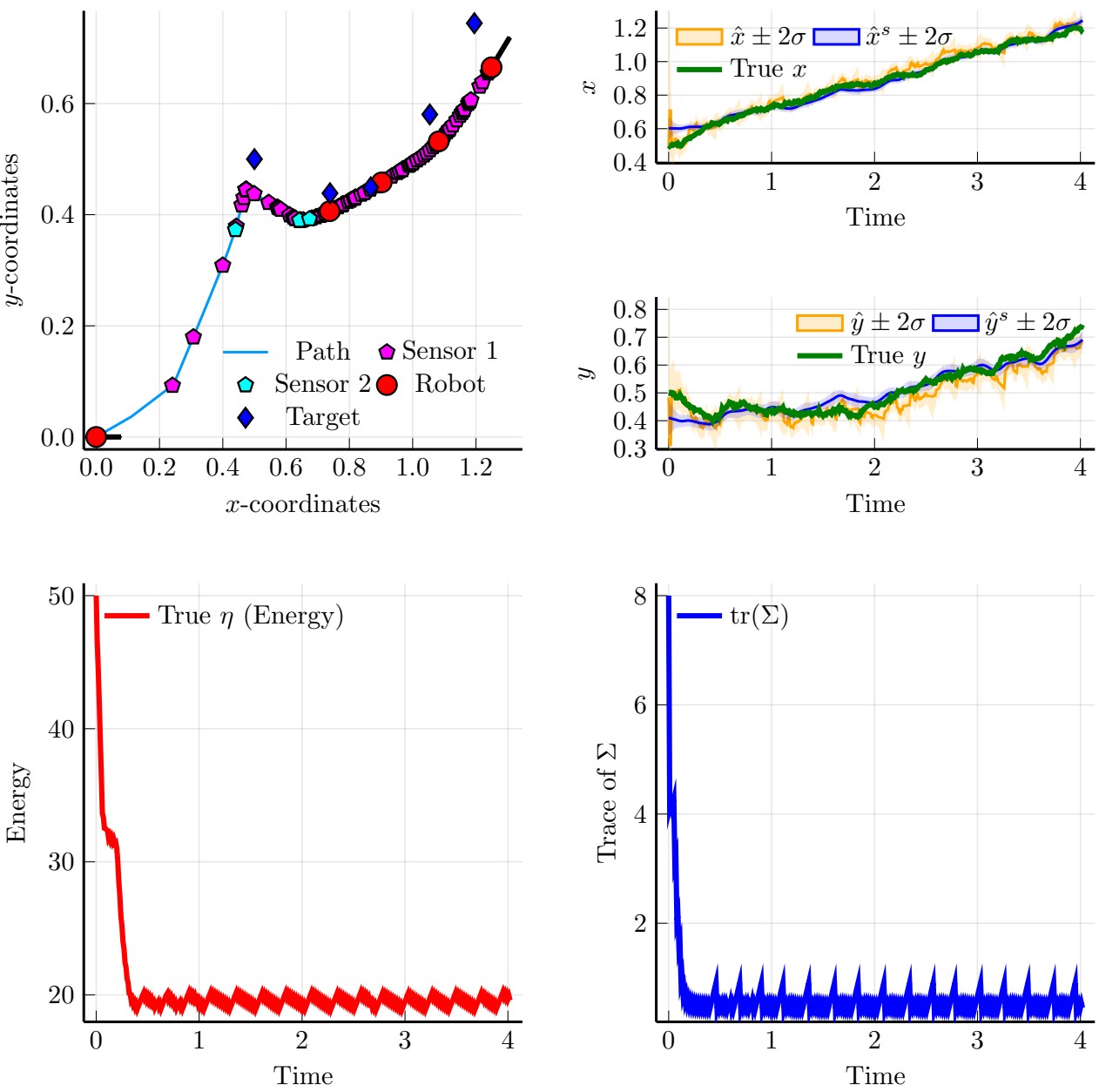

*Figure 6.* Target Tracking example utilizing an RH approach. The KF position estimates are $\hat{x}, \hat{y}$ and the RTS position estimates are $\hat{x}^s, \hat{y}^s$. See an animation of the results on `https://github.com/MOHAMMADZAHD93/When2measureKF`.

the model (27) which was used for simulation. The running cost for each OCP is

$$\mathcal{L}(\hat{\xi}, \hat{\Sigma}, u, \lambda, \varepsilon) = w_\Sigma \operatorname{tr}(\hat{\Sigma}) + w_\lambda \lambda^\top \lambda + w_u u^\top u$$

where $w_\Sigma \geq w_\lambda \geq w_u \geq 0$, and the terminal cost being the distance between the robot and the predicted location of the target according to the model in (28). The running constraints $\mathcal{C}$ encodes the input bounds $\underline{u} \leq_e u \leq_e \bar{u}$, $\lambda \geq_e 0$, and an energy lower bound $\hat{\eta} \geq c_\eta$ with $c_\eta \geq 0$. The terminal constraint $\mathcal{C}_T$ encodes a tighter lower bound on energy to ensure feasibility for the next optimization. The results are illustrated in Figure G. The results in Figure G show that the RH implementation manages to make the robot obtain close estimates to the target's position, which enables the robot to track

it closely. Additionally, we can see that the energy drops to a lower value close to the terminal constraint on energy with measurement-induced oscillations. We also see that the same thing happens to the covariance matrix of the CD-KF.

## H. Experiments for Nonconcave/Nonconvex Auxiliary Dynamics

In this section, we will solve the OCP in (12) for several cases in which we modify the auxiliary dynamics for the robot example in Section 9 and the water sensors in Appendix E to violate the concavity/convexity assumption 5.1. For the robot example in Section 9, we consider three different cases in which we modify equation (17) to be

$$\frac{d\zeta_s}{dt} = \sum_{i=1}^{N_s} \gamma_i \exp\big(-r_\zeta \|p_r - p_p\|^2\big) \left(\frac{1}{2} + \frac{1}{2}\psi_\zeta(\zeta_s)\right) \delta_{t_i^s}, \tag{29}$$

where $\psi_\zeta(\zeta_s) = \frac{1}{1+\exp(-\zeta_s)}$ for the first case, $\psi_\zeta(\zeta_s) = \frac{1}{(1+\exp(-\zeta_s))^3}$ for the second and third case (the degradation cost doubles as the degradation state for each sensor increases). The difference in the third case is that we modify the energy dynamics in equation (14) to be

$$\frac{d\eta}{dt} = c_e \psi_{\eta_1}(\eta) \exp\big(r_e \|p_r - p_e\|^2\big) - c_u v^2 - c_u \omega^2 - \sum_{s=1}^{2} \sum_{i=1}^{N_s} c_s \psi_{\eta_2}(\eta) \, \delta_{t_i^s}, \tag{30}$$

where $\psi_{\eta_1}(\eta) = \frac{1+\exp(-(\eta-20))}{2+\exp(-(\eta-20))}$ (charging efficiency is halfed as the stored energy increases), and $\psi_{\eta_2}(\eta) = 1 + \exp(-\eta^2/5^2)$ (low stored energy increases the energy costs). For the example of the water sensors in Appendix E, we consider a case in which fouling equation in (26) is modified to be

$$\frac{d\xi_p^s}{dt} = -\big(\alpha_s^f + \gamma_s^f u_s(t)\big) \big(\xi_p^s(t)\big)^3 + \sum_{i=1}^{N_s} \rho_s^f \delta_{t_i^s}, \ s \in \{1, 2\}. \tag{31}$$

The results for the different cases are shown in Figure 7, Figure 8, Figure 9, and Figure 10. The figures for robot cases show that we can still obtain satisfactory results even if the dynamics for the auxiliary states violate assumption 5.1. The same can also be said for the results of the modified water fouling example in Figure 10, except for the fouling state of the second sensor, where it can be seen that the planned fouling becomes different from the true one towards the end. Nevertheless, the planned and true covariance matrix $\Sigma$ for all of these examples are similar. These empirical examples support the idea in Remark 7.2 that our method can still work in situations where the dynamics of the perturbed auxiliary state have a local near-linear behaviour.

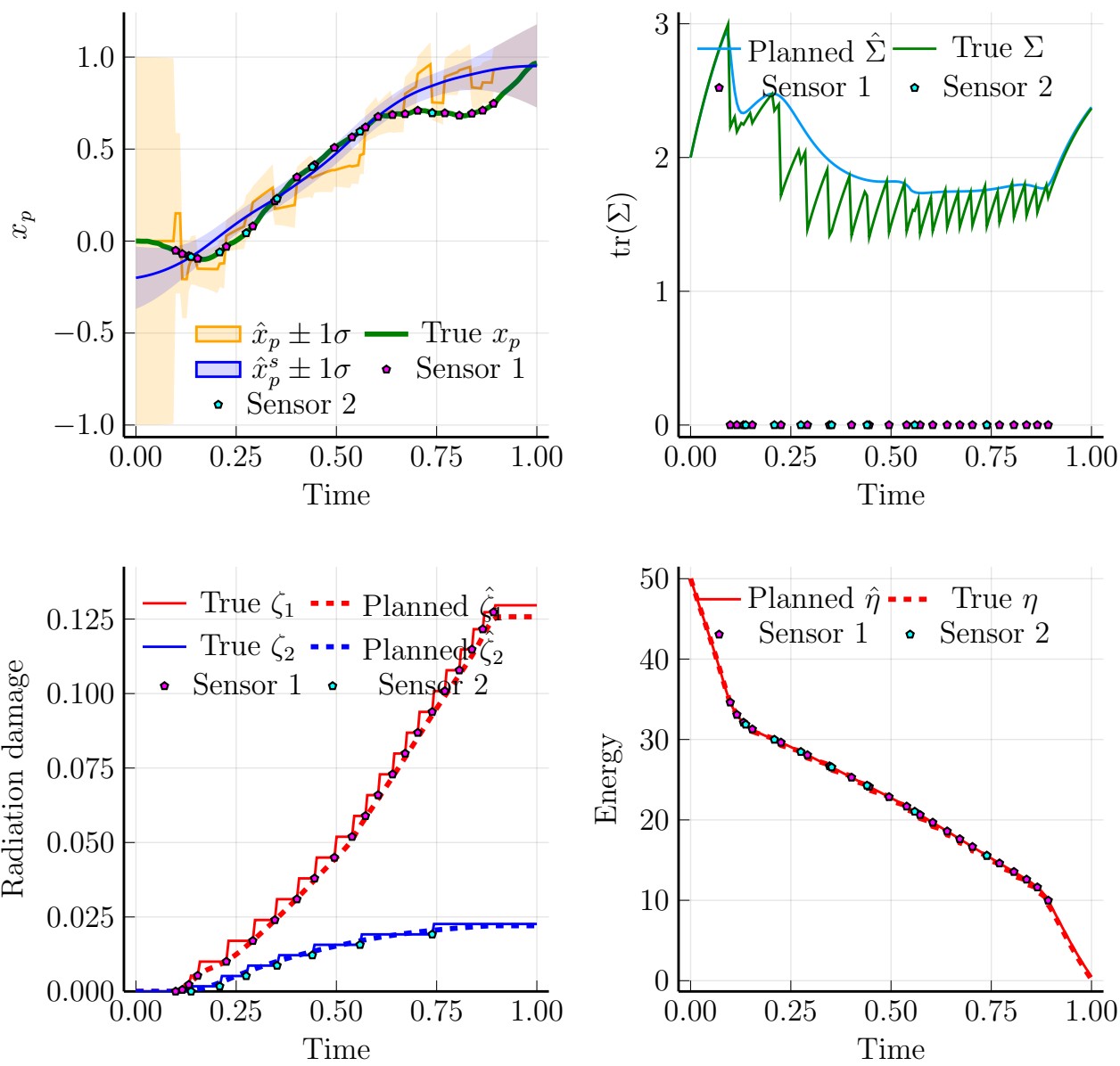

*Figure 7.* The first modified (nonconvex/nonconcave) case for the robot example in Section 9. The KF estimate is $\hat{x}_p$ and the RTS estimate is $\hat{x}_p^s$.

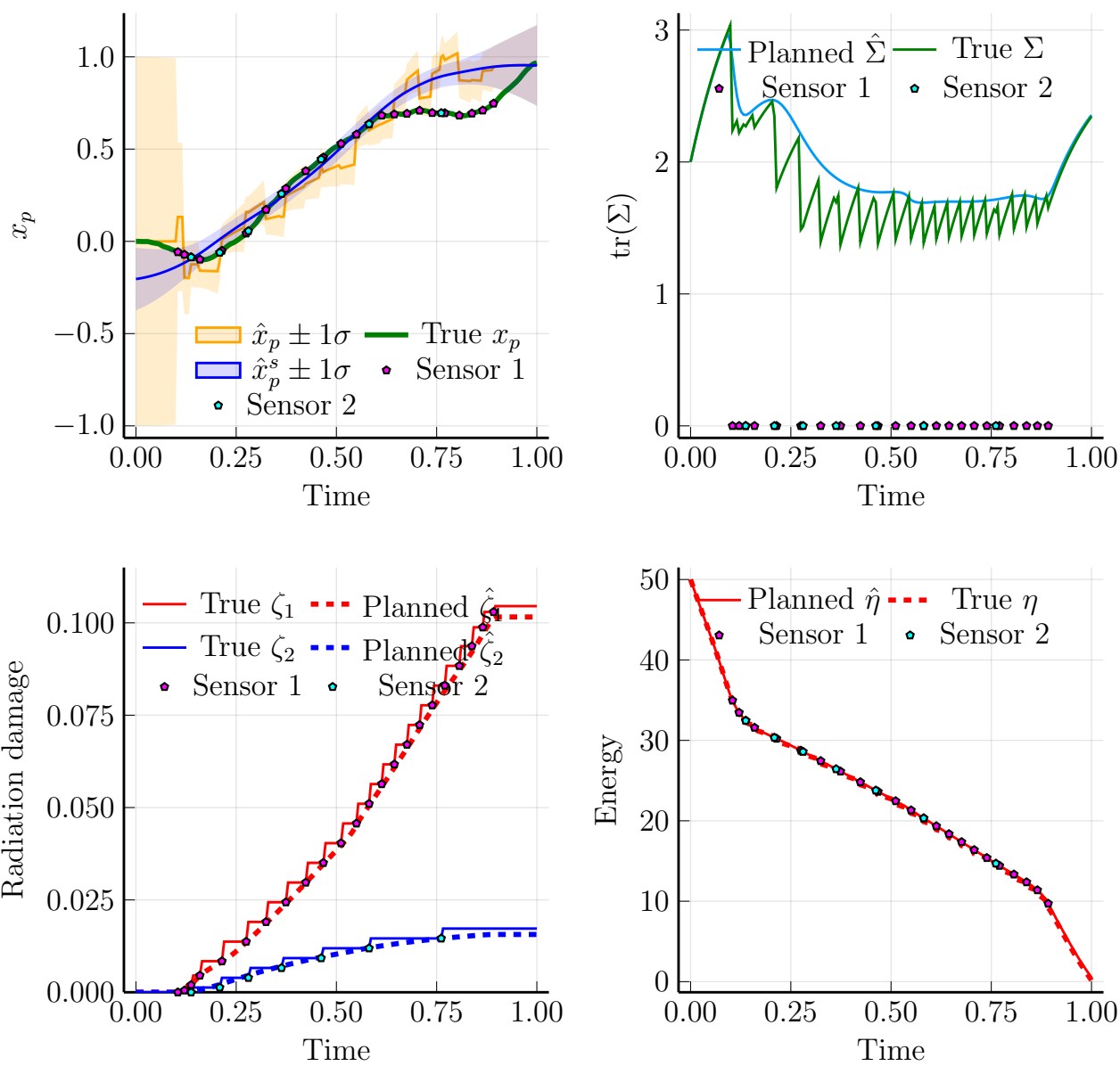

*Figure 8.* The second modified (nonconvex/nonconcave) case for the robot example in Section 9. The KF estimate is $\hat{x}_p$ and the RTS estimate is $\hat{x}_p^s$.

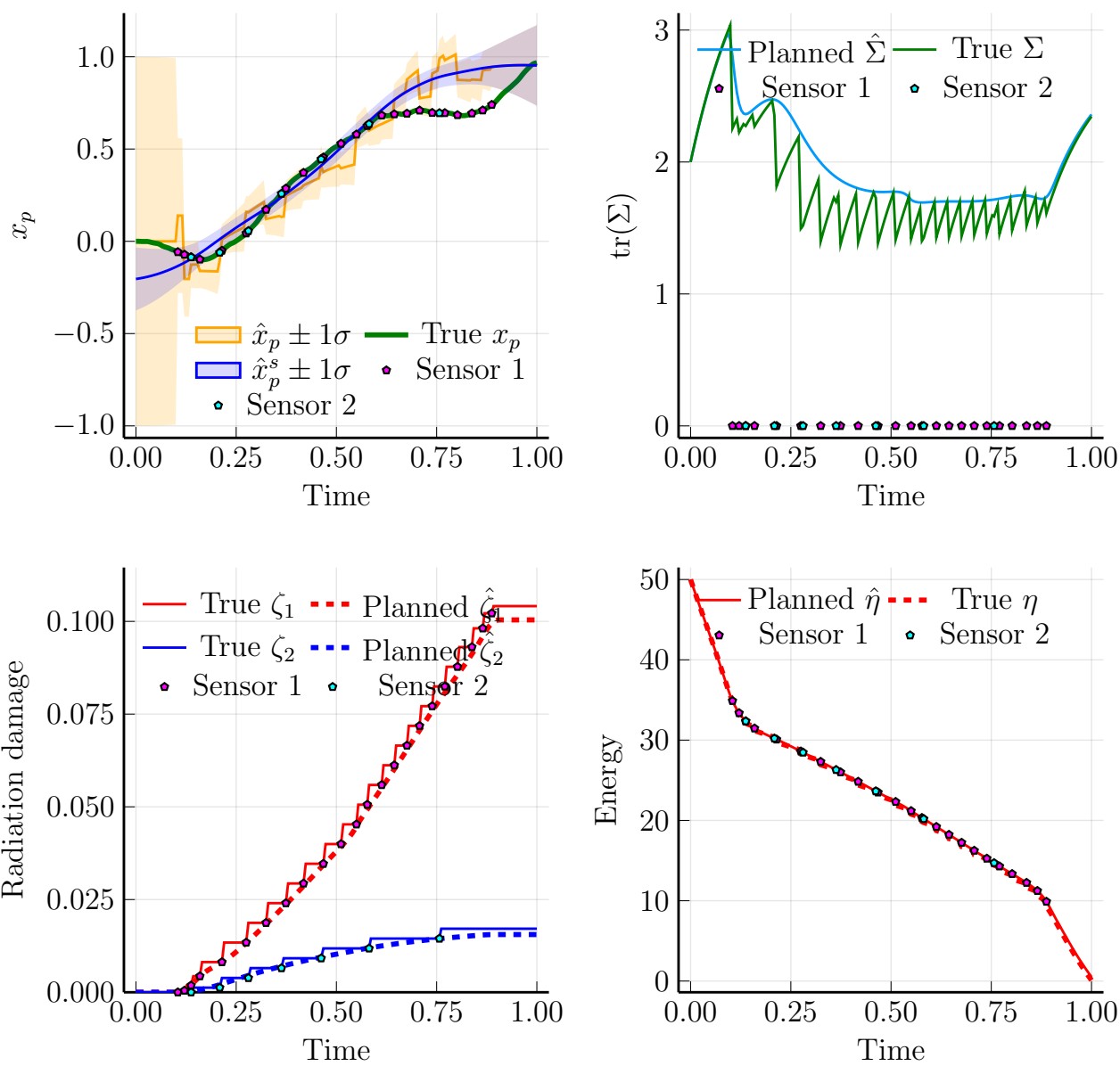

Figure 9. The third modified (nonconvex/nonconcave) case for the robot example in Section 9. The KF estimate is $\hat{x}_p$ and the RTS estimate is $\hat{x}_p^s$.

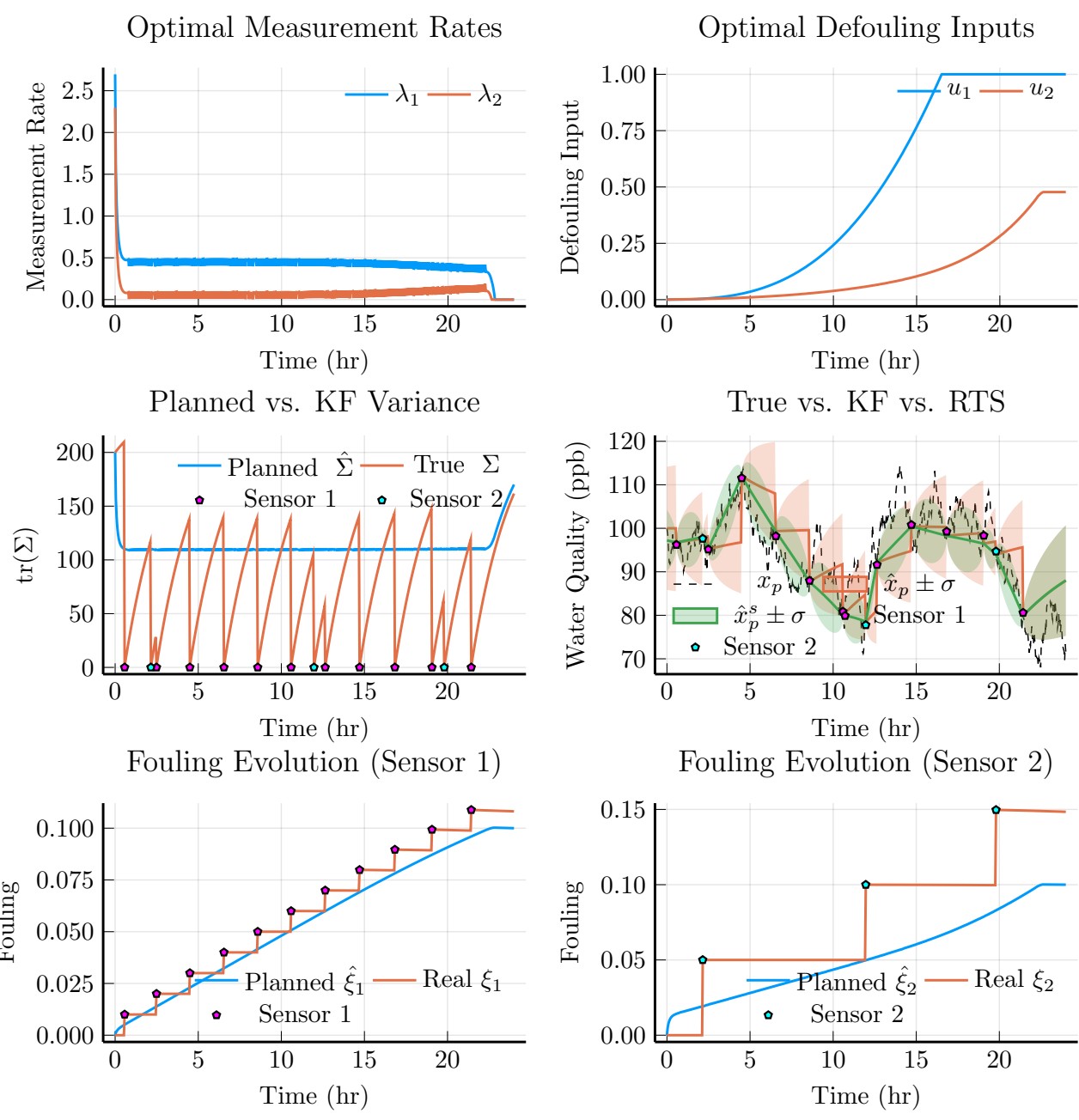

Figure 10. The modified water fouling example in Appendix E (nonconvex/nonconcave). The KF estimate is $\hat{x}_p$ and the RTS estimate is $\hat{x}_p^s$.

## I. State-Space Realization for Gaussian Processes

A one-dimensional GP with a covariance kernel whose power spectral density is rational admits an exact representation as an SSM (Hartikainen & Särkkä, 2010; Särkkä & Hartikainen, 2012; Todescato et al., 2020). In this framework, forward filtering via the CD-KF combined with backward RTS smoothing computes the exact posterior mean and covariance of GP regression (Hartikainen & Särkkä, 2010). The CD-KF and RTS algorithms achieve this with $\mathcal{O}(N)$ computational complexity for $N$ measurements (in contrast to $\mathcal{O}(N^3)$ with the traditional Gaussian Process regression), thus enabling scalable inference for large datasets.

For linear-Gaussian models, the smoothing covariance matrix is always dominated by the filtering covariance in the Loewner order. Consequently, the OCP in (12) inherently controls both the filtering and smoothing uncertainties. This means our framework for sensor scheduling with auxiliary dynamics in this paper can be used in connection with efficient GP regression.

For kernels with non-rational spectral densities (e.g., the squared-exponential kernel), approximations can be constructed using rational function expansions such as Padé approximants (Todescato et al., 2020). These approximations will yield finite-dimensional SSM representations.

## J. Unicycle Model

For a unicycle model of a robot, we have the state

$$\xi_p = \begin{bmatrix} p_r \\ \theta \end{bmatrix}, \quad \text{where} \quad p_r = \begin{bmatrix} p_{r_1} \\ p_{r_2} \end{bmatrix}$$

represents the planar position with respect to a global frame, and $\theta$ is the heading angle. The linear (heading) velocity is $v$ and the angular velocity is $\omega$. The kinematic equations are then

$$\dot{\xi}_p = \begin{bmatrix} \dot{p}_{r_1} \\ \dot{p}_{r_2} \\ \dot{\theta} \end{bmatrix} = \begin{bmatrix} v\cos(\theta) \\ v\sin(\theta) \\ \omega \end{bmatrix}.$$

