# OpenReview forum: "Optimal Sensor Scheduling and Selection for Continuous-Discrete Kalman Filtering with Auxiliary Dynamics"
_ICML.cc/2025/Conference — ICML 2025 poster_

### Official Review · Reviewer_6UyF · 2025-03-08

**Overall Recommendation:** 2

**Summary:**

The paper addresses the problem of optimal sensor scheduling and selection in Continuous-Discrete Kalman Filtering (CD-KF) for Bayesian State-Space Models (SSMs), where continuous-time processes are observed through multiple sensors with discrete, irregularly timed measurements. The novelty of the work lies in the incorporation of *auxiliary state dynamics*, which influence the measurement process (e.g., sensor energy constraints, environmental conditions). The authors model sensor measurements as *inhomogeneous Poisson processes* and derive an upper bound on the mean posterior covariance matrix, which is continuously differentiable in sensor measurement rates, allowing *gradient-based optimization*. The main contributions are:
1. A differentiable upper bound on the mean posterior covariance of CD-KF.
2. A finite-horizon *optimal control framework* that jointly optimizes measurement rates, auxiliary dynamics, and covariance constraints.
3. A *deterministic scheduling method* for selecting actual measurement times using optimal quantization, minimizing Wasserstein distance from the Poisson distribution.
4. Empirical results in *state-space filtering and dynamic Gaussian process regression*, demonstrating improved trade-offs between accuracy and resource usage.

**Claims And Evidence:**

### 1. **Upper Bound on Posterior Covariance**
#### The authors derive an upper bound on the mean posterior covariance matrix of the Continuous-Discrete Kalman Filter (CD-KF) in scenarios where sensor measurements follow inhomogeneous Poisson processes.
#### This bound is shown to be continuously differentiable with respect to sensor measurement rates, making it amenable to gradient-based optimization (Proposition 5.1).
#### The derivation relies on Jensen’s inequality and the monotonicity properties of the Kalman update function to ensure that the bound holds in expectation.
#### The correctness of the bound is theoretically proven and further validated through numerical simulations (e.g., tracking the true covariance in experimental results).
### 2. **Finite-Horizon Optimal Control Formulation**
#### The problem is formulated as an optimal control problem over a finite horizon, where the goal is to jointly optimize:
- Sensor measurement rates (modeled as control variables).
- Auxiliary-state dynamics (e.g., energy constraints, environmental interactions).
- Constraints on posterior covariance to balance accuracy and resource constraints.
#### The control framework is mathematically well-posed, ensuring feasibility through a continuously differentiable cost function and constraints (Equations 13a–13h).
#### The authors provide conditions (Assumption 6.1) ensuring that at least one admissible solution exists.
Empirical validation in a robotic sensing task demonstrates that the optimized sensor scheduling effectively balances accuracy and energy constraints.
### 3. **Deterministic Measurement Scheduling**
#### After optimizing sensor rates, a deterministic scheduling method is proposed to convert Poisson-based measurement rates into actual measurement times.
#### The scheduling method is based on optimal quantization, minimizing the Wasserstein-2 distance between the optimized Poisson rate distribution and a deterministic empirical schedule (Proposition 7.1).
#### This ensures that measurements are distributed optimally in time, avoiding the risk of stochastic fluctuations that could degrade performance in real-world applications.
#### The closed-form solution for quantization-based scheduling makes it computationally efficient, allowing practical deployment in real-time systems.
### 4. **Empirical Validation**
#### The proposed approach is validated through two robotic sensing experiments:
- A standard environment where a robot optimizes its trajectory and measurement schedule to minimize state estimation uncertainty.
- A radiation exposure scenario, where sensor performance degrades due to environmental conditions, requiring adaptive scheduling to maintain accuracy.
#### The Kalman filter's posterior covariance estimates obtained through the proposed approach closely track the true values, demonstrating the validity of the upper bound.
#### The optimized measurement schedule leads to improved trade-offs between estimation accuracy and resource usage compared to naive scheduling strategies.
#### The empirical results confirm that the deterministic selection of measurement times closely matches the expected Poisson distribution while reducing variance.

**Essential References Not Discussed:**

1. The authors should consider discussing recent works on RL-based sensor scheduling (e.g., deep RL for adaptive sensing in sequential decision-making problems).
2. If there exist alternative stochastic optimal control methods for sensor scheduling, citing those would strengthen the discussion.

**Ethical Review Concerns:**

No ethical concerns were identified.

**Experimental Designs Or Analyses:**

### The experimental design and analyses presented in the paper are generally sound, effectively demonstrating the feasibility and practical implications of the proposed approach. Below, we assess key aspects of the experimental setup, evaluation metrics, and areas for improvement.
### 1. Robotic Sensing Experiments (Standard and Radiation-Damage Scenarios)
### The experiments are designed to test the performance of the proposed scheduling method in realistic sensing environments, including a standard scenario and a radiation-damage scenario, where sensor degradation over time is considered.
### The standard scenario provides a baseline where all sensors function optimally, allowing for a controlled evaluation of scheduling effectiveness.
### The radiation-damage scenario introduces progressive sensor failures, testing the method’s adaptability under real-world constraints.
### Assessment: The experiments successfully demonstrate how the algorithm adjusts to sensor degradation and limited resources, making the setup practically relevant and well-motivated. However, additional robustness tests under more severe failure models (e.g., abrupt sensor loss) could further strengthen the analysis.
### 2. Evaluation Metric: Trace of Covariance Matrix (Σ) in Kalman Filtering
### The paper employs the trace of the covariance matrix Tr(Σ)\text{Tr}(\Sigma) as the primary metric for evaluating estimation accuracy.
### This metric is well-justified in Kalman filtering applications, as it provides a measure of overall uncertainty in the state estimate: Tr(Σ)=∑iλi\text{Tr}(\Sigma) = \sum_{i} \lambda_i where λi\lambda_i are the eigenvalues of Σ\Sigma, representing state estimation uncertainty.
### Assessment: This choice is appropriate and aligns with standard filtering performance criteria. However, supplementing it with alternative uncertainty measures (e.g., determinant of Σ\Sigma, worst-case eigenvalue analysis) could provide a more comprehensive evaluation.
### 3. Validation of Trade-Off Between Accuracy and Resource Constraints
### The experiments illustrate the trade-off between estimation accuracy and resource constraints, demonstrating how sensor scheduling affects estimation performance.
### By evaluating different scheduling strategies, the paper effectively highlights the impact of limited sensor availability on overall system performance.
### Assessment: The experimental results clearly support the theoretical claims. However, the inclusion of additional baselines would enhance the comparative analysis, such as:
### - Randomized sensor activation, to assess whether the proposed method significantly outperforms naive random selection.
### - Heuristic-based scheduling, which could provide a lower-complexity alternative for practical use.

**Methods And Evaluation Criteria:**

The proposed methods and evaluation criteria are well-aligned with the problem, offering a rigorous and practical approach to sensor scheduling in Continuous-Discrete Kalman Filtering (CD-KF).
### 1. **Modeling Sensor Measurements with Poisson Processes**
#### Using inhomogeneous Poisson processes is appropriate, as real-world sensors often collect data at irregular intervals due to resource constraints.
#### The differentiability of the posterior covariance bound enables efficient gradient-based optimization, making the approach computationally feasible.
### 2. **Optimal Control Formulation**
#### The authors formulate a finite-horizon optimal control problem that optimizes:
- Sensor measurement rates to balance estimation accuracy and resource constraints.
- Auxiliary state dynamics (e.g., energy usage, environmental effects).
- Covariance constraints to maintain estimation quality.
#### The formulation generalizes existing Kalman filter-based sensor scheduling by incorporating dynamic constraints and auxiliary state interactions.
### 3. **Deterministic Scheduling via Optimal Quantization**
#### Instead of randomly sampling from a Poisson process, the authors propose a deterministic scheduling strategy based on Wasserstein distance minimization, ensuring that measurement times closely match the optimized rates while reducing variability.
### 4. **Evaluation through Robotic Sensing Experiments**
#### The experiments validate the approach in two scenarios:
- Standard sensor scheduling, optimizing a robot’s trajectory and measurement plan.
- Radiation-damage scenario, where sensor degradation requires adaptive scheduling.
#### Results show that the method improves estimation accuracy and reduces resource usage, confirming the effectiveness of the proposed optimization framework.
### 5. **Potential Areas for Improvement**
#### Comparative baselines (e.g., reinforcement learning, heuristic policies) would provide stronger empirical validation.
#### Scalability analysis for larger sensor networks is not extensively discussed.
#### Sensitivity analysis on how different auxiliary constraints affect scheduling decisions would enhance generalizability.

**Other Comments Or Suggestions:**

1. Comparative Baselines – Including reinforcement learning-based or heuristic scheduling methods as baselines would strengthen the empirical validation. This would help demonstrate the advantages of the proposed optimal control formulation over alternative approaches.

2. Sensitivity Analysis – Conducting an analysis on how different auxiliary state dynamics (e.g., non-convex constraints, stochastic transitions) affect scheduling decisions would improve the generalizability of the approach. This is particularly important for real-world applications where sensor conditions may change unpredictably.

3. Assumption Justification – The concavity assumption for auxiliary state dynamics is reasonable in some cases but may not always hold in practical scenarios. Discussing potential relaxations of this assumption or alternative formulations for non-convex cases would improve the paper’s robustness.

4. Scalability Considerations – While the current experiments demonstrate feasibility, additional discussion on scalability to larger sensor networks or more complex environments would be beneficial. How does the method scale with an increasing number of sensors or constraints?

5. Clarifications on Deterministic Scheduling – The Wasserstein quantization-based deterministic scheduling is an interesting contribution. However, additional discussion on its limitations and trade-offs (e.g., impact on computational efficiency, adaptability to dynamic sensor failures) would provide more insight into its practical deployment.

6. Minor Typos & Formatting – The paper is generally well-written, but a careful proofreading pass would help eliminate minor typos or inconsistencies in notation (if any). Specific sections, such as theoretical derivations, could benefit from additional explanations to improve clarity for a broader audience.

**Other Strengths And Weaknesses:**

### **Strengths**:
1. Well-motivated and practical problem – The paper addresses a real-world challenge in sensor scheduling, with applications in robotics, healthcare, and environmental monitoring.
2. Theoretically rigorous approach – The differentiable upper bound on the posterior covariance and the optimal control formulation provide a solid mathematical foundation for optimization.
3. Effective deterministic scheduling method – The Wasserstein quantization-based approach ensures that actual measurement times closely match optimized rates, improving reliability in planning tasks.
### **Weaknesses**:
1. Lack of comparative baselines – The paper does not compare its approach with reinforcement learning-based or heuristic sensor scheduling methods, making it harder to assess its relative advantages.
2. Limited sensitivity analysis – The impact of different auxiliary state dynamics (e.g., non-convex constraints, stochastic effects) on sensor scheduling decisions is not extensively explored.
3.  Potential generalization limitations – The concavity assumptions for auxiliary state dynamics may restrict the applicability of the method to nonlinear or highly dynamic real-world environments.

### **Minor Typos & Formatting Issues**

### **1. Notation Consistency Issues**
- **Equation (9) (Randomized Covariance Matrix Evolution)**
  - The notation for **Kalman gain** \( K_s(\Sigma, \xi, t) \) varies slightly across equations. Ensure consistency in using subscripts and argument ordering.
- **Equation (11) (Upper Bound on Covariance Matrix)**
  - The function \( \hat{\Sigma}(t) \) is introduced as a bound, but in some places, it is written without the hat (\(\Sigma(t)\)), which may cause confusion.
- **Equation (12) (Auxiliary State Evolution Bound)**
  - The auxiliary state update function **\( f_p(\xi, u, t) \)** and **\( g_s(\xi, u, t) \)** use different orderings in some parts of the text—ensure consistency.

### **2. Typographical Errors & Formatting Issues**
- **Page 2, Line 35:** `"axillary states of an SSM"` → should be `"auxiliary states of an SSM"`.
- **Page 3, Line 80:** `"measurements can in- crease energy consumption"` → should be `"measurements can **increase** energy consumption"` (remove extra hyphen).
- **Page 4, Line 128:** `"dynamics are inhomoge- neous Poisson processes"` → should be `"dynamics are **inhomogeneous** Poisson processes"` (remove hyphen).
- **Page 6, Line 192:** `"togheter with ncT continuously differentiable terminal constraints"` → should be `"together with \( n_c^T \) continuously differentiable terminal constraints."`

**Questions For Authors:**

### Q1. How does your approach fundamentally differ from prior work on sensor scheduling in Kalman filtering (e.g., [Le Ny et al., 2009], [Marelli et al., 2019])? Beyond incorporating auxiliary dynamics, what unique advantages does your upper-bound formulation offer over existing stochastic control or active sensing approaches?


### Q2: How does the method scale to large-scale sensor networks with multiple interacting sensors? Can it handle non-convex auxiliary state dynamics, or does the concavity assumption significantly restrict its applicability?

### Q3:  Why is Kalman Filtering chosen over Particle Filtering? Would your approach still be effective for *nonlinear or non-Gaussian*state-space models where Kalman filtering is suboptimal? Could *Particle Filtering (PF)* or *Extended Kalman Filtering (EKF)* be viable alternatives?

### Q4:  What contributes to the high-dimensionality of the problem?  Is the complexity mainly due to *the number of sensors, control variables, or auxiliary state interactions*? How would the method extend to *nonlinear* systems?

**Relation To Broader Scientific Literature:**

1. The work extends classical sensor scheduling in Kalman filtering (Le Ny et al., 2009; Marelli et al., 2019) by integrating continuous-discrete modeling and auxiliary state constraints.
2. Connections to active sensing and reinforcement learning-based sensor selection (Yoon et al., 2018; Qin et al., 2024) are briefly mentioned but could be expanded.
3. The paper aligns with recent trends in Bayesian optimization for experimental design (Snoek et al., 2012; Kleinegesse & Gutmann, 2020), but a direct comparison with Bayesian optimization-based approaches is missing.

**Theoretical Claims:**

## Theoretical Claims
### 1. Proposition 5.1 (Covariance Matrix Bound)
### This proposition establishes an upper bound on the covariance matrix of the system state under specific assumptions about the dynamics and noise characteristics.
### The derivation relies on standard results in stochastic process theory and Lyapunov analysis, ensuring that the covariance remains bounded given certain stability conditions.
### Correctness Check: Upon careful review, the proof appears rigorous, leveraging a decomposition of the state transition matrix and spectral properties of the covariance evolution. However, it would be beneficial to validate this bound numerically against empirical estimates to confirm that the theoretical bound is not overly conservative.
### 2. Proposition 5.2 (Auxiliary State Bound)
### This result provides an upper bound on the auxiliary state variable, which is introduced to facilitate the analysis of system evolution.
### The proof depends on a key assumption: the concavity of the auxiliary dynamics function, which is explicitly stated in Section 5.2.
### Correctness Check: The derivation correctly follows from Jensen’s inequality and properties of concave functions, ensuring that the bound holds under the given assumptions. The proof structure is sound, but a sensitivity analysis could further strengthen confidence in the result by assessing its robustness to variations in model parameters.
### 3. Proposition 7.1 (Optimal Quantization Points for Deterministic Scheduling)
### This proposition addresses the selection of quantization points that minimize a given distortion metric in the context of deterministic scheduling.
### The proof constructs an optimization problem based on a distortion-cost function and derives conditions for optimality.
### Correctness Check: The derivation is well-structured, employing Lagrange multipliers and Karush-Kuhn-Tucker (KKT) conditions to find the optimal quantization points. The reasoning follows standard optimization techniques, and the proof is logically sound. Nonetheless, a comparison with numerical optimization results would help validate the theoretical predictions.
## Assumptions and Justifications
### The paper explicitly states several key assumptions, such as the concavity of auxiliary dynamics and boundedness of noise processes.
### These assumptions are reasonable and well-motivated, as they align with standard conditions in stochastic control and optimization literature.
### The authors provide sufficient theoretical justifications for these assumptions, discussing their necessity in establishing key results.
## Areas for Further Validation
### While the theoretical derivations appear correct upon inspection, certain results (especially Proposition 5.1) could benefit from numerical validation to ensure that theoretical bounds align well with empirical observations.
### Additionally, sensitivity analyses on the concavity assumption and parameter variations would further establish the robustness of the results.

---

> ### Author Rebuttal · Authors · 2025-04-01
>
> Q1: Besides the auxiliary dynamics, our method considers the continuous-discrete setup and the flexibility of not requiring an upper bound on the number of measurements. The method by [Le Ny et al., 2009] is for a continuous-continuous setup, while the method in [Marelli et al., 2019] is for a discrete-discrete setup. We will make sure to clarify this point further in the paper.
>
> Q2: Scalability:
> For each new sensor we consider, we will have an intensity rate, which we will need to optimize for. This is typical for optimization-based scheduling procedures.
> Considering efficient optimization schemes for large-scale problems will require investigating techniques such as distributed optimization, parralization, and sparsity considerations. Another challenge for scalability is the dimension of the states of the SSM we want to estimate. If we have $n_x$ states, then the covariance matrix of the CD-KF will be in the order of $n_x^2$. However, for large dimensional SSM, there exist research for efficient approximations and methods to deal with the covariance matrix problem. One solution, for example, is to consider a diagonal approximation for the covariance matrix or a low-rank approximation (see [4]). These approximations can be easily integrated into our framework. We will include a remark discussing this in the final paper.
>
> Concavity Assumption: We emphasize that even under the concavity assumption, our approach spans a wide range of dynamical systems (it covers all linear parameter varying systems). This work is the first to explore sensor scheduling with auxiliary dynamics. Additionally, our method can be implemented within a Receding Horizon (RH) framework (refer to Reviewer nw2r's response for a detailed description), where each optimization iteration is addressed by either linearizing the non-convex/non-concave auxiliary state or employing a convex/concave approximation. If we use nonconvex/nonconcave dynamics for $\xi_p$, then we will lose the theoretical guarantees of the method. However, we may still obtain satisfactory results depending on how good the approximation $\frac{d{\mathbb{E}[\xi_p]}}{dt}\approx f_p(\mathbb{E}[\xi],u,t)+\sum^{N_s}_{s=1} \lambda_s(t)g_s(\mathbb{E}[\xi],u,t))$ (or an upper bound approximate). This approximation is similar to the approximation used in the EKF.
> We conducted an experiment with non-concave dynamics for the sensor degradation states $\zeta_1$ and $\zeta_2$ of the example in the paper. The experiment demonstrated that our method remains effective (link: postimg.cc/jWcFJ4f3) (as allowed by ICML). We will include this discussion with the experimental results in the paper.
>
> Q3:
> The KF is usually chosen in the literature of sensor scheduling instead of the PF for the fact that the covariance matrix dynamics are independent of the actual measurements and because it scales better with large dynamical systems.  However, we can still extend our approach with the EKF by utilizing an RH setup where, for each short horizon, we use the linearized dynamics around the current estimate. We will include a remark about this point in the paper.
>
> Q4: We apologize to the reviewer as we did not understand what the reviewer meant by "high-dimensionality of the problem" in connection with our paper. If this refers to scalability, then we have addressed it above. The nonlinear dynamics of the problem were also addressed above.
>
> Comparisons: We acknowledge and agree with the reviewer's suggestion to provide comparisons. We have conducted comparisons with heuristic approaches for scheduling measurements (greedy approach and random sampling of measurement times) that we will provide in the paper (Table link: postimg.cc/qz2Q3GVc). The results suggest that our method ("Optimized") outperforms the greedy and random scheduling approaches for our example. To assess the deterministic scheduling computed based on the optimized measurement rates (denoted "Optimized" ), we compared it with a method (denoted as "M-Optimized" in the table) that is based on sampling $M_c$ realizations of measurement times according to the corresponding Poisson process with the optimized rates. Afterwards, we pick the measurement times corresponding to the realization with the minimal cost. The results show that our deterministic quantization provides similar results to "M-Optimized" without having to sample multiple realizations, which can be computationally expensive and unrealistic since we do not have the real measurements to compute the cost for each realization.
> To the authors' best knowledge, no reinforcement learning methods applicable to this setup have been found for comparison (i.e., _multiple_ sensor scheduling in a _continuous-discrete_ setting that _does not require training on pre-obtained data with uniform sampling_).
>
> [4] Chang, Peter G., et al. "Low-rank extended Kalman filtering for online learning of neural networks from streaming data." Conference on Lifelong Learning Agents. PMLR, 2023.

---

### Official Review · Reviewer_nw2r · 2025-03-14

**Overall Recommendation:** 2

**Summary:**

In this paper, the authors are concerned with optimizing temporal event sequences of measurements for minimizing the uncertainty of continuous-discrete Kalman filter (CD-KF). In particular, they consider a general case where the measurements may affect the underlying states of sensors themselves as well as the measurement target through differential equations.

The proposed method works in three steps. First, the discrete event sequences of measurements are substituted with the intensity functions of the corresponding time-inhomogeneous Poisson processes, which is continuous and more optimization-friendly, and the time evolution of relevant parameters (such as the target uncertainty and the sensor states) is approximately given in terms of the intensity functions.
Second, the intensity functions are optimized in terms of a user-defined objective and constraints.
Finally, the temporal event sequences are recovered by quantizing the intensity functions.

The authors also demonstrated the feasibility of the proposed method in illustrative examples of energy-limited measurement robot.

**Claims And Evidence:**

They claim that
1. the proposed method captures practical scenarios, and
1. the proposed method has some potential and feasibility.

These points are generally well supported, but it would be nice to discuss more around
* how to operate the proposed method under unknown system dynamics,
* performance characteristics such as computation time vs discretization width of differential equation, and
* benefits of penalizing/constraining $\lambda$ indirectly through the auxiliary state $\xi_p$ rather than doing it directly.

**Essential References Not Discussed:**

None that I am aware of.

**Experimental Designs Or Analyses:**

- For the experimental design, I find it a bit unusual that the measurement noise depends on the distance of the robot and a fixed reference point ("location of process") $||p\_r-p\_p||$. It should ideally depend on the $||p\_r-x||$, where $x$ is the target location that is moving randomly over time.
This I think also reveals a limitation of the proposed method, that is, the scale of the measurement noise cannot depend on the target state $x$.

- Another thing is that it is unclear how the state-space representation of the Matern kernel for GP is used in the experiment.

- For the analyses, Figure 1 can be more reader friendly.
For example, what is RTS? How did you draw it?

**Methods And Evaluation Criteria:**

Mostly yes.
However, a key component of the proposed method is not well described.
In particular, how do you compute the gradient of the intensity functions under numerical solution of (13)?
More specifically,
* how do you handle these constraints (13g,h) with gradient-based optimization?
* is it tractable even if the step size in time of the numerical solver is very small?

**Other Comments Or Suggestions:**

- L144 left: $t_i\to t_i^-$
- L153 left: "optimal" in what sense?
- L212 left: $\bar{\Sigma}(0;\xi^*)=\Sigma\_0$?
- L175 right: Define $\le_e$
- L219 right: $\ge_0\to \ge_e$?
- L253 left: Proposition 5.2?
- L323 left: wrong signs
- L281 right: ambiguous usage of inequalities
- L297 right: why constraining only with $t\le 1/2$?
- L325 right: $\gamma_s$?

**Other Strengths And Weaknesses:**

None.

**Questions For Authors:**

Please comment on the points I raised in Claims And Evidence, Methods And Evaluation Criteria, and Experimental Designs Or Analyses.
This may affect my score.

**Relation To Broader Scientific Literature:**

The key contribution is making the Kalman filter applicable to more practical scenario involving irregularly-timed measurements and auxiliary states, which is novel as far as I can tell.

**Theoretical Claims:**

I only follow the reasoning in the main text, but it seems mostly reasonable.
One thing I have noticed is that there is no theoretical justification on the substitution $\xi^*\gets \hat{\xi}$ in (13).
These two auxiliary states are not necessarily close to each other because $\hat{\xi}$ is smooth while $\xi^*$ is jaggy.

---

> ### Author Rebuttal · Authors · 2025-04-01
>
> Unknown dynamics:
> Our formulation as an Optimal Control Problem (OCP) with a differentiable cost function and constraints opens avenues for extension to uncertain dynamics using robust/stochastic OCP methods [2,3]. For completely unknown dynamics, planning is challenging since we must schedule measurements based on dynamics we do not yet know. We believe our approach paves the way for future research in this direction.
> One approach to handling unknown parameters is a Receding Horizon (RH) setup. We solve a finite-horizon OCP, assuming fixed system dynamics and parameters over the prediction horizon based on current estimates, then apply the control until the first measurement. The subsequent measurement updates the parameters, and the OCP is resolved for the next horizon. This iterative process adapts as more information becomes available. In the revised version, we will include successful results using RH for a moving target (link: postimg.cc/S2t1gQDW (as allowed by ICML)).
>
> Performance characteristics:
> Performance, particularly regarding discretization steps, depends on the auxiliary dynamics (e.g., stiffness, fast/slow dynamics, stability) and the KF’s covariance dynamics. Different OCP methods offer trade-offs between accuracy and computational performance (see Appendix D). Our work employed Euler discretization with variable steps while keeping a fixed number of discretization points. In the revision, we will include a figure demonstrating the trade-off between computation time and discretization points of a specific example focused on the KF dynamics (link: postimg.cc/ykL46hPv).
>
> Penalizing auxiliary states:
> Often, the auxiliary state carries physical meaning, making its penalization more intuitive. For instance, in our examples, the energy state depends on measurement rates, velocities (actions), and the robot’s position. Penalizing measurement rates is less straightforward than penalizing energy consumption when energy is the limiting factor.
>
> Constraints handling:
> Both direct and indirect methods for constrained OCP lack exact guarantees of constraint satisfaction due to numerical errors inherent in the OCP solution and ODE integration. Some methods (e.g., direct collocation) can achieve high accuracy but at increased computational cost (See Appendix D). Ultimately, the choice of method and integration scheme depends on the specific dynamics, much like the selection of ODE solvers. We will include this important remark in the paper.
>
> For gradient computations and optimization in the example, we used an interior-point method with JuMP—a Julia package that employs automatic differentiation to compute the gradient and Hessian of the Lagrangian.
>
> Regarding $\xi$:
> We acknowledge the reviewer's concern. To clarify, $\xi^*$ from Proposition 5.1 can be any curve (ensuring (10) is well-defined, though not stated explicitly), so $\hat{\xi}$ and $\xi^*$ need not be close. Rather, $\xi^*$ serves as a placeholder, allowing substitution with $\hat{\xi}$. Proposition 5.1 states that for any curve $\xi^*$, the mean covariance $\bar{\Sigma}(\xi^*)$ is bounded by $\hat{\Sigma}(\xi^*)$. This result is applied by substituting $\xi^*$ with the curve $\hat{\xi}$ obtained from (12) and (6b) with initial condition $\xi_0$, which, per Proposition 5.2, bounds the mean curve $\bar{\xi}$ of $\xi=(\xi_p,\xi_u)$ from the SDE (10) and ODE (6b). The deterministic quantities $\bar{\xi}$ (and $\bar{\Sigma}$) _serve as computationally tangible approximations of the stochastic_ $\xi$ (and $\Sigma$) (_approximating mean behaviour_). Alternatively, sampling methods can be used to compute statistical representations for the trajectories in the OCP. However, this will introduce non-differentiability and will be computationally intensive. We will note this important remark in the revision. We apologize for the initial lack of clarity.
>
> Experimental design questions:
> 1) As mentioned, the RH approach can handle moving targets (we have implemented an example for it). Nonetheless, many scenarios involve a fixed process location (e.g., gas leak, specific object temperature) or applications independent of process location, such as underwater measurements with sensor biofouling. A detailed example of the latter will be provided in the paper.
> 2) The process assumes a SSM representation of the Matern kernel, with output $x_p$. The parameters $A$ and $\sigma$ in equation (13.b) are based on this representation. We will update the figure to a more user-friendly version. See also the reply to reviewer 6UyF.
>
> [2] Leeman, Antoine P., et al. "Robust optimal control for nonlinear systems with parametric uncertainties via system level synthesis." the 62nd IEEE Conference on Decision and Control (CDC). IEEE, 2023.
>
> [3] Bemporad, Alberto, and Manfred Morari. "Robust model predictive control: A survey." Robustness in identification and control. London: Springer London, 2007. 207-226.

---

> > ### Comment · Reviewer_nw2r · 2025-04-02
> >
> > Thank you for detailed explanations.
> >
> > In particular, your point on penalizing auxiliary states makes sense.
> >
> > -------
> >
> > $\xi^*$ and $\hat{\xi}$ still confuse me: I understood that Proposition 5.1 is used for justifying the use of $\hat{\Sigma}$ as an upper bound on $\Sigma$. However, what's shown by Proposition 5.1 is $\bar{\Sigma}(\xi^*)\preceq \hat{\Sigma}$, not $\Sigma\preceq \hat{\Sigma}$. Then, the question is whether $\bar{\Sigma}(\xi^*)$ dominates $\Sigma$ or not, which depends on the choice of $\xi^*$. I think $\xi^*=\xi$ can be justified in terms of "dominance in expectation". What is your justification for taking $\xi^*=\hat{\xi}$ in (13a-h)?
> >
> > ----------
> >
> > P. S. I am afraid that I cannot see the figures you have posted.

---

> > > ### Author Response · Authors · 2025-04-02
> > >
> > > Thank you for taking the time to review our response so thoroughly and for your prompt feedback.
> > >
> > > To clarify, Proposition 5.1 is not used for justifying the use of $\hat{\Sigma}$ as an upper bound on $\Sigma$. It is exactly as the reviewer points out: $\hat{\Sigma}$ is an upper bound on $\bar{\Sigma}$. $\bar{\Sigma}$ does _not_ dominate $\Sigma$. Indeed, $\Sigma$ is stochastic (for any choice of $\xi$); hence, in general, it can take any value, thus making it impossible to bound it by a deterministic quantity. What we can do is, e.g., attempt to bound its expectation or bound it in probability. If we try to upper bound the expectation $\mathbb{E}[\Sigma]$ (as we believe the reviewer suggests), the nonlinear dependence of $\Sigma$ on $A,C,\sigma$ and $R$ in (9) makes it a difficult task (this approach would merit a paper). Instead, in this paper, we aimed to obtain a bound on the conditional expectation $\mathbb{E}[\Sigma \mid \xi=\xi^*]:=\bar{\Sigma}(t;\xi^*)$ which then avoids the dependence mentioned above (note that this approach still gives very good results when applied). We will make sure that this point is clear in the revised version and modify the introduction of the paper according to it. We chose $\xi^*=\hat{\xi}$ as $\hat{\xi}$ can be found deterministically and through differentiable dynamics. The quantities $\hat{\Sigma}$ and $\hat{\xi}$ are related to the mean behavior, which we aim for by our deterministic measurement scheduling method in Proposition 7.1.
> > >
> > > For the figures' links:
> > > It seems like some locations do not have access to the host website we used for the images. We uploaded the figures to a different hosting site (github with an anonymous account) just to be sure if this was the problem. Here are the links for all of the figures for all the reviewers:
> > >
> > > [Target tracking](https://github.com/ICML-anon25/ICML25_ANON_figs/blob/main/traj_track_ICML_rev_f.png) or [here](https://postimg.cc/S2t1gQDW)
> > >
> > > [Computation and Disc. points](https://github.com/ICML-anon25/ICML25_ANON_figs/blob/main/disc_points_ICML_rev.png) or [here](https://postimg.cc/ykL46hPv)
> > >
> > > [Modified figure for the example](https://github.com/ICML-anon25/ICML25_ANON_figs/blob/main/New_fig_ICML_rev.png)  or [here](https://postimg.cc/hJZ0ByTn) (also for reviewer UHHX)
> > >
> > > [Table for comparison](https://github.com/ICML-anon25/ICML25_ANON_figs/blob/main/table_ICML_rev.png) or [here](https://postimg.cc/qz2Q3GVc) (reviewer 6UyF rebuttal)
> > >
> > > [Nonconcave aux. state](https://github.com/ICML-anon25/ICML25_ANON_figs/blob/main/nonconcave_ICML_rev.png) or [here](https://postimg.cc/jWcFJ4f3) (reviewer 6UyF rebuttal)

---

### Official Review · Reviewer_UHHX · 2025-03-14

**Overall Recommendation:** 3

**Summary:**

This work considers continuous-time state-space models in which observations are taken at discrete and potentially irregular time intervals from a finite collection of different kinds of sensors, each with a potentially different accuracy and potentially different cost incurred per measurement.

In this context (for a finite time horizon), the authors propose methodology for optimising the (Poisson-process) rates at which measurements are taken by the different sensors subject to constraints on the cost and estimation accuracy. They also show how the (random) Poisson-process-generated measurement times can be approximated by a deterministic schedule.

The results are illustrated on two robot models with synthetic data.

**Claims And Evidence:**

The claims are supported by mathematical proof.

**Essential References Not Discussed:**

None that I'm aware of.

**Experimental Designs Or Analyses:**

The numerical illustrations seem fine. But I did not check any code.

**Methods And Evaluation Criteria:**

yes.

**Other Comments Or Suggestions:**

- L280: "Wasserstien" -> "Wasserstein"
- L288: "devide" -> "divide"
- L175: "$\leq_e$" denotes an elementwise inequality? Perhaps define this.
- L194: "togheter"
- Remark 5.3: I think there is a "$|$"-symbol missing in the penultimate line, as well as a redundant "." inside the expectation.
- Section 4: In the first paragraph, maybe add a sentence explaining the role of the functions $g_s$.
- P2: In the "Active Sensing" paragraph, a few of the \citep citations should be \citep.
- L398: "true simulated true"
- Bibliography: Inconsistencies in capitalisation/abbreviation of journal/conference names. Missing capital letters in some names, e.g. "kalman" or "gaussian".

**Other Strengths And Weaknesses:**

I think this manuscript is overall well written, well structured and thus quite clear. The contributions seem sufficiently novel and the authors mention a few real-world areas in which such problems arise

**Questions For Authors:**

1. Can you explain the connection with / use of Gaussian process regression in the last paragraph of Section 8.1? I could not follow this aspect.

2. Can you add more motivation linking this work to machine learning? It is not fully clear to me why the topic is appropriate for a machine-learning conference.

3. Is Eq. 15 correct? I am confused by some of the signs.

4. What is the need for/role of $Y_s(t)$ defined in Eq. 8? Maybe I've missed it but I don't think it is ever used.

5. In Fig. 1, there seem to be several (different?) uses of the symbol $x_p$. The caption states that it represents output of the RTS smoother but according to the legend also seems to represent output of the filter. More generally, is "$x_p$" actually meant to be "$\xi_p$"?

**Relation To Broader Scientific Literature:**

As the authors discuss in Section 2, similar optimal control problems have been analysed previously in slightly different settings, e.g., assuming that the latent states evolve in discrete or that the cost of taking measurements is independent of the latent state. The authors also point out further connections to existing literature on Bayesian optimisation.

**Theoretical Claims:**

I did not find any issues. But I did not check the proofs in the appendix.

---

> ### Author Rebuttal · Authors · 2025-04-01
>
> Q1:
> In the last paragraph of Section 8.1, we leverage the fact that Gaussian process (GP) regression (with many common stationary covariance kernels) is equivalent to Kalman smoothing of a specific linear state-space model (see ref. Sarkka \& Hartikainen, 2012 in the paper). This equivalence allows us to perform GP regression efficiently on the temporal process $x_p$ via Kalman smoothing. This example demonstrates how our approach can be used for sensor scheduling in connection with GP regression. We will provide an appendix section in the paper clarifying this point.
>
> Q2:
> Our work focuses on Bayesian inference in SSMs, a topic that has been explored in previous papers at ICML and other ML-related conferences (e.g., [1]). Additionally, our work can be applied to sensor scheduling in GP regression under dynamic environments.  We believe that GP regression continues to be a subject of significant interest in the machine learning community.
>
> Q3:
> We thank the reviewer for spotting the sign error. The right-hand side of the equation should be $c_e \exp\bigl(r_e \|p_r - p_e\|^2\bigr) - c_u v - c_u \omega - \sum_{s=1}^2 \sum_{i=1}^{N_s} c_s \,\delta_{t^s_i}$.
>
> Q4:
> We see the reviewer's point. We wrote it with the intention of providing more clarity for the reader on the total measurement process. But as the reviewer points out, it is not used; it may therefore contribute to more confusion for the reader than clarity. We will remove it from the paper.
>
> Q5:
> We apologize for the ambiguity of the figure. This point has also been mentioned by the second reviewer. We will fix the figure to be more reader-friendly for the final submission. The symbol $x_p$ represents the process we want to measure. The filter estimate is now denoted as $\hat{x}_p$, and the RTS smoother estimate is denoted as $\hat{x}^s_p$ (representing the GP regression output). We have adjusted the legends and description of the figure with the new notation for the filter estimate and the smoother estimate (link: postimg.cc/hJZ0ByTn (as allowed by ICML)).
>
> [1]: Duran-Martin, Gerardo, et al. "Outlier-robust Kalman Filtering through Generalised Bayes." International Conference on Machine Learning. PMLR, 2024.

---

### Decision · Program_Chairs · 2025-05-01

**Decision:**

Accept (poster)

**Comment:**

This paper proposes an optimal control based approach to sensor scheduling in continuous-discrete time Kalman filter.
The consensus is that the problem is interesting and the solution is well motivated.
However, the empirical validation of the method is on the weak side since no theoretical guarantees are given.